

# Two-species hardcore reversible cellular automaton: matrix ansatz for dynamics and nonequilibrium stationary state

**Marko Medenjak[1], Vladislav Popkov[2,3,4], Tomaž Prosen[2], Eric Ragoucy[5] and Matthieu Vanicat[2⋆]**

**1** Institut de Physique Théorique Philippe Meyer, Ecole Normale Supérieure, PSL University, Sorbonne Universités, CNRS, 75005 Paris, France
**2** Faculty of Mathematics and Physics, University of Ljubljana, Jadranska 19, SI-1000 Ljubljana, Slovenia
**3** Institut für Theoretische Physik, Universität zu Köln Zülpicher Str. 77, 50937 Köln
**4** Bergisches Universität Wuppertal, Gauss Str. 20, D-42097 Wuppertal
**5** Laboratoire d'Annecy-le-Vieux de Physique Théorique LAPTh, Université Grenoble Alpes, USMB, CNRS, F-74000 Annecy

⋆ matthieu.vanicat@fmf.uni-lj.si

## Abstract

In this paper we study the statistical properties of a reversible cellular automaton in two out-of-equilibrium settings. In the first part we consider two instances of the initial value problem, corresponding to the inhomogeneous quench and the local quench. Our main result is an exact matrix product expression of the time evolution of the probability distribution, which we use to determine the time evolution of the density profiles analytically. In the second part we study the model on a finite lattice coupled with stochastic boundaries. Once again we derive an exact matrix product expression of the stationary distribution, as well as the particle current and density profiles in the stationary state. The exact expressions reveal the existence of different phases with either ballistic or diffusive transport depending on the boundary parameters.



# 1 Introduction

The determination of the macroscopic behavior of a system from its microscopic dynamics is a major challenge of statistical mechanics. One would like, for instance, to derive the hydro-dynamic equations which govern the flow of conserved physical quantities (energy, electric charge, mass, etc.) starting from the knowledge of the microscopic interactions between the elementary constituents of the system. The problem is in general mathematically very difficult to address, because of the huge number of degrees of freedom of the many-body systems. In the last years the study of integrable systems and the development of the generalized hydro-dynamics framework lead to the posibility of a precise large scale description of the dynamics of the charge and current profiles of all conserved quantities [1,2]. Nonetheless, the results of the generalized hydrodynamics rely on uncontrolled assumptions which can typically be corroborated only by the numerical checks.

It is therefore important to investigate other theoretical techniques, which will enable us to study the non-equilibrium behavior of many-body systems analytically. Here we develop the matrix ansatz method to compute the time evolution of the probability distributions and to obtain the stationary states. The matrix ansatz technique has been first introduced in the context of the Totally Asymmetric Simple Exclusion Process, to express the steady state [3],

and has attracted a lot of attention since then in the context of continuous-time [4–11] and discrete-time [12–14] classical stochastic processes and also in the context of open quantum systems [15, 16]. In order to further develop the method, it thus appears primordial to design very simple models in which it is possible to perform analytical computations. One of the simplest class of many-body dynamical systems one can think of are the cellular automata. These are deterministic systems defined on a discrete space-time lattice with a discrete (typically finite) set of states at each space-time point. In particular, reversible cellular automata can be considered as minimal models for Hamiltonian (conservative) dynamics in the context of statistical mechanics [17, 18]. In the present paper we investigate exact matrix product solutions for a specific 2-species reversible cellular automata introduced in [19, 20] which may be considered as a model of a charged hardcore gas in one dimension. From the physical perspective the model is interesting since it exhibits a crossover between ballistic and diffusive transport.

Here we study the model in three different settings. The first one describes an inhomogeneous (bipartite) quench obtained by joining together two equilibrium distributions at different charge densities. The second one describes a local defect quench obtained by perturbing an equilibrium distribution on a single site. This setup can also be understood as a time propagation of a local observable. Less general quench setups were studied already in [19, 20] using a different method. The last setting describes a boundary driving obtained by stochastic interactions of the system with the particle reservoirs.

The objective of the paper is twofold. Firstly, from a more mathematical perspective, we provide an exact matrix product expressions of the time-evolution of probability distributions using a unified framework, which relies on the algebraic "cancellation mechanism". It is one of the first examples of the explicit time-dependent matrix product ansatz, beyond the recent results [21, 22] which rely on the soliton counting. We expect that our framework can be generalized to more complicated systems. We also construct the exact matrix product expression of the stationary state of a stochastic boundary driven system. Secondly, from a more physical perspective, we provide new results associated with a phase transition between ballistic and diffusive phases of a boundary driven model.

In section 2 we present the dynamics of the model and introduce the different physical settings that we are interested in, namely the two quench protocols and the stochastic boundary driving. Then in section 3 (and in section 4) we study in details the inhomogeneous quench protocol (and the local quench protocol respectively). We provide an exact matrix product expression of the time-evolution of the system and we use it to compute analytically the density profiles. Finally, in section 5 we provide an exact matrix product expression of the stationary state of the boundary driven system. We compute analytically the particle currents and the density profiles for any size of the system. This allows us to identify the transition between the ballistic phase and the diffusive phase and obtain a complete phase diagram of the boundary driven system.

# 2 A two-component reversible cellular automaton

## 2.1 Definition of the models

We are interested in different models that share the same bulk dynamics, but with different boundary conditions and/or different initial conditions. In the first two cases, we will consider a periodic lattice with two different initial conditions (inhomogeneous and local quenches), while in the last case, we will be on a finite segment connected to two reservoirs. In the first setting, we will be interested in the propagation cone emerging from the origin, with the assumption that the propagation cone is not affected by the periodicity (*i.e.* we will always

consider the case where the time $t < L/4$).

In order to give a precise mathematical definition of the models and to study them, we need to introduce some notations. The models are defined on a finite lattice comprising $L$ sites labeled by an integer $i \in [1, L]$. Each site of the lattice is either empty or can carry a particle of species (or charge) $+$ or $-$. We denote by $\tau_i^t$ the local occupation variable at site $i$ and at time $t$. More precisely $\tau_i^t = 0$ (respectively $\tau_i^t = +$, $\tau_i^t = -$) if there is a vacancy (respectively a particle of charge $+$, a particle of charge $-$) on site $i$ at time $t$. The configuration of the lattice at time $t$ is thus given by the $L$-uplet $\underline{\tau}^t = (\tau_1^t, \tau_2^t, \ldots, \tau_L^t)$.

Our aim is to study the statistical properties of the models. For this purpose it is useful to introduce probability distributions. For $\tau_1, \ldots, \tau_L \in \{0, +, -\}$, we denote by $p_{\tau_1, \ldots, \tau_L}^t$ the probability that $\underline{\tau}^t = (\tau_1, \tau_2, \ldots, \tau_L)$. It will be convenient to encompass the probabilities of all configurations in a single vector

$$\boldsymbol{p}(t) = \sum_{\tau_i \in \{0, +, -\}} p_{\tau_1, \ldots, \tau_L}^t e_{\tau_1} \otimes e_{\tau_2} \otimes \cdots \otimes e_{\tau_L}, \tag{1}$$

where

$$e_0 = \begin{pmatrix} 1 \\ 0 \\ 0 \end{pmatrix}, \qquad e_+ = \begin{pmatrix} 0 \\ 1 \\ 0 \end{pmatrix}, \qquad e_- = \begin{pmatrix} 0 \\ 0 \\ 1 \end{pmatrix} \tag{2}$$

are the elementary basis vectors of $\mathbb{C}^3$.

We now present the dynamics of the models. As already mentioned, the models that we are interested in share the same deterministic bulk dynamics. It is defined locally by updating a pair of neighboring sites $(\tau_i^{t+1}, \tau_{i+1}^{t+1}) = \phi(\tau_i^t, \tau_{i+1}^t)$ with the following rules

$$\phi(0, \tau) = (\tau, 0), \qquad \phi(\tau, 0) = (0, \tau), \qquad \phi(\epsilon, \epsilon') = (\epsilon, \epsilon'), \tag{3}$$

where $\tau \in \{0, +, -\}$ and $\epsilon, \epsilon' \in \{+, -\}$. In words, the particles are propagating freely through the vacancies and there is a hard-core interaction between the particles, so that they cannot exchange their positions. We are considering a two-step discrete-time dynamics. During the first time step only the pairs of odd-even sites are updated whereas during the second time step only the pairs of even-odd sites are updated. This can be summarized as $(\tau_i^{t+1}, \tau_{i+1}^{t+1}) = \phi(\tau_i^t, \tau_{i+1}^t)$ for $t - i$ even.

For the periodic lattice, the complete dynamics is fully defined by the previous rules and the fact that the indices should be considered modulo the size of the system $L$. Note that in this case the number of sites should be even.

For the open case, we are considering a lattice with an odd number of sites, so that at each time-step either the first or the last site of the lattice is singled out. When $t$ is even we update $\tau_1^t$ with the following stochastic rule: $\tau_1^{t+1}$ is equal to $0$ with probability $\alpha_0$, to $+$ with probability $\alpha_+$, and to $-$ with probability $\alpha_-$ (where $\alpha_0 + \alpha_+ + \alpha_- = 1$). All other sites are updated accordingly to the bulk dynamics. Similarly, if $t$ is odd, we update $\tau_L^t$ with the following stochastic rule: $\tau_L^{t+1}$ is equal to $0$ with probability $\beta_0$, to $+$ with probability $\beta_+$, and to $-$ with probability $\beta_-$ (where $\beta_0 + \beta_+ + \beta_- = 1$), and the remaining sites accordingly to the bulk dynamics.

In both periodic and open lattice settings the time evolution of the probability vector $\boldsymbol{p}(t)$ obeys a master equation of the following form

$$\boldsymbol{p}(t+1) = \begin{cases} \mathbb{U}^e \boldsymbol{p}(t), & t \text{ even}, \\ \mathbb{U}^o \boldsymbol{p}(t), & t \text{ odd}, \end{cases} \tag{4}$$

where the two operators $\mathbb{U}^e$ and $\mathbb{U}^o$ correspond to the two different time-steps. In the first setting with periodic boundary conditions, we have

$$\mathbb{U}^e = \prod_{k=1}^{\frac{L}{2}} U_{2k,2k+1} \qquad \text{and} \qquad \mathbb{U}^o = \prod_{k=1}^{\frac{L}{2}} U_{2k-1,2k}. \tag{5}$$

The subscripts indicate on which sites of the lattice the operators are acting non-trivially. The operator $U$ is a $9 \times 9$ matrix acting on two tensor space components and encoding the local update rule on two neighboring sites. It reads

$$U = \begin{pmatrix} 1 & 0 & 0 & 0 & 0 & 0 & 0 & 0 & 0 \\ 0 & 0 & 0 & 1 & 0 & 0 & 0 & 0 & 0 \\ 0 & 0 & 0 & 0 & 0 & 0 & 1 & 0 & 0 \\ 0 & 1 & 0 & 0 & 0 & 0 & 0 & 0 & 0 \\ 0 & 0 & 0 & 0 & 1 & 0 & 0 & 0 & 0 \\ 0 & 0 & 0 & 0 & 0 & 1 & 0 & 0 & 0 \\ 0 & 0 & 1 & 0 & 0 & 0 & 0 & 0 & 0 \\ 0 & 0 & 0 & 0 & 0 & 0 & 0 & 1 & 0 \\ 0 & 0 & 0 & 0 & 0 & 0 & 0 & 0 & 1 \end{pmatrix}. \tag{6}$$

Note that it satisfies the braid relation

$$U_{12} U_{23} U_{12} = U_{23} U_{12} U_{23}, \tag{7}$$

leading to an $R$-matrix

$$R(z) = P \frac{\mathbb{I} + z U}{z + 1}, \tag{8}$$

where $P$ is the permutation operator $P(u \otimes v) \equiv v \otimes u$ acting on the tensor space $\mathbb{C}^3 \otimes \mathbb{C}^3$. The $R$-matrix obeys the Yang-Baxter equation (with additive spectral parameter $z$), the unitarity condition $R(z)R(-z) = \mathbb{I}$ and $R(0) = PU$.

From this $R$-matrix and any $\lambda \in \mathbb{C}$, one builds the transfer matrix (we remind that $L$ is even for periodic boundary conditions)

$$t_\lambda(z) = tr_0\Big(R_{0L}(z)R_{0,L-1}(z+\lambda)R_{0,L-2}(z)R_{0,L-3}(z+\lambda)\cdots R_{02}(z)R_{01}(z+\lambda)\Big), \tag{9}$$

which commutes for different values of the spectral parameter. Then, standard calculation leads to

$$t_\lambda(-\lambda)^{-1} t_\lambda(0) = V_{12}^+ V_{34}^+ \cdots V_{L-1,L}^+ V_{L-2,L-1}^+ V_{L-4,L-3}^+ \cdots V_{23}^+ V_{L,1}^+, \tag{10}$$

where $V^\pm = \frac{\mathbb{I} \pm \lambda U}{1 \pm \lambda}$. We have used the relations $(V^-)^{-1} = V^+$ and $U^2 = \mathbb{I}$. From the property $\lim_{\lambda \to \infty} V^\pm = U$, one gets

$$\lim_{\lambda \to \infty} \left( t_\lambda(-\lambda)^{-1} t_\lambda(0) \right) = \mathbb{U}^o \mathbb{U}^e,$$

ensuring the integrability of the model with periodic boundary condition. $\mathbb{U} = \mathbb{U}^o \mathbb{U}^e$ commutes with the transfer matrix

$$t_\infty(z) = tr_0\Big(R_{0L}(z) \check{U}_{0,L-1} R_{0,L-2}(z) \check{U}_{0,L-3} \cdots R_{02}(z) \check{U}_{01}\Big), \quad \text{where} \quad \check{U} = PU. \tag{11}$$

In the second, open lattice setting with stochastic boundary driving, we have

$$\mathbb{U}^o = \left( \prod_{k=1}^{\frac{L-1}{2}} U_{2k-1,2k} \right) \overline{B}_L \qquad \text{and} \qquad \mathbb{U}^e = B_1 \left( \prod_{k=1}^{\frac{L-1}{2}} U_{2k,2k+1} \right). \tag{12}$$

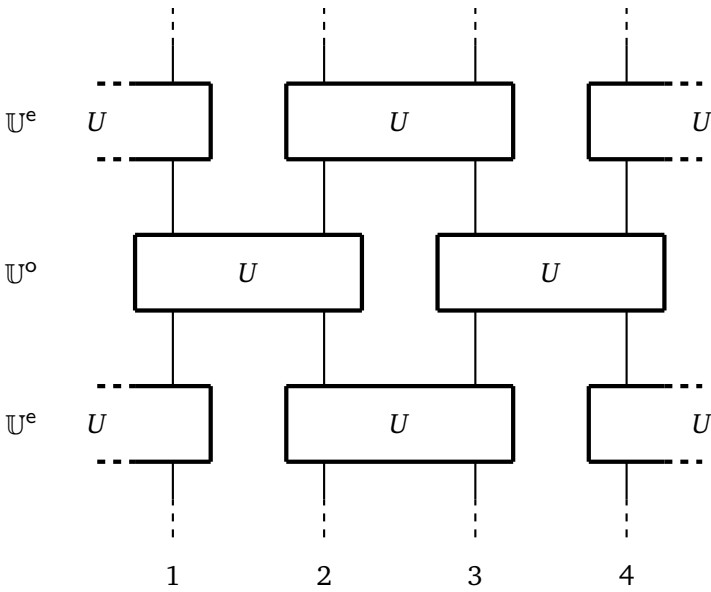

Figure 1: Pictorial representation of the discrete time dynamics for periodic boundary conditions. Note that the time flows upward.

Once again the subscripts indicate on which sites of the lattice the operators are acting non-trivially. The boundary operators $B$ and $\overline{B}$ are $3 \times 3$ matrices acting on a single tensor space component and encoding the stochastic rules on the first and last sites. They read

$$B = \begin{pmatrix} \alpha_0 & \alpha_0 & \alpha_0 \\ \alpha_+ & \alpha_+ & \alpha_+ \\ \alpha_- & \alpha_- & \alpha_- \end{pmatrix}, \qquad \overline{B} = \begin{pmatrix} \beta_0 & \beta_0 & \beta_0 \\ \beta_+ & \beta_+ & \beta_+ \\ \beta_- & \beta_- & \beta_- \end{pmatrix}. \tag{13}$$

Note that they can not be obtained from solutions to the reflection equation associated with the $R$-matrix given in (8).

## 2.2 Separable states

We present here the building blocks of the quench protocols studied in sections 3 and 4. The bulk system dynamics allows for the existence of a class of spatially homogeneous current-carrying states invariant under the system dynamics. They are defined for a system with periodic boundary conditions (the system size $L$ is then even), or alternatively for a system with (fine-tuned) open boundary conditions ($L$ is then odd). In both cases the respective stationary probability distribution obtains a factorized form,

$$p_{s_1 s_2 s_3 \dots} = \alpha_{s_1} \beta_{s_2} \alpha_{s_3} \dots , \tag{14}$$

$$p'_{s_1 s_2 s_3 \dots} = \beta_{s_1} \alpha_{s_2} \beta_{s_3} \dots , \tag{15}$$

where $\boldsymbol{p}$ and $\boldsymbol{p}'$ represent probability vectors over all particle configurations after odd and even times, respectively. For this reason we shall call them separable states. It is easy to see that $\mathbb{U}^{\mathrm{o}}\boldsymbol{p} = \boldsymbol{p}'$, $\mathbb{U}^{\mathrm{e}}\boldsymbol{p}' = \boldsymbol{p}$, if and only if

$$U \begin{pmatrix} \alpha_0 \\ \alpha_+ \\ \alpha_- \end{pmatrix} \otimes \begin{pmatrix} \beta_0 \\ \beta_+ \\ \beta_- \end{pmatrix} = \begin{pmatrix} \beta_0 \\ \beta_+ \\ \beta_- \end{pmatrix} \otimes \begin{pmatrix} \alpha_0 \\ \alpha_+ \\ \alpha_- \end{pmatrix}, \tag{16}$$

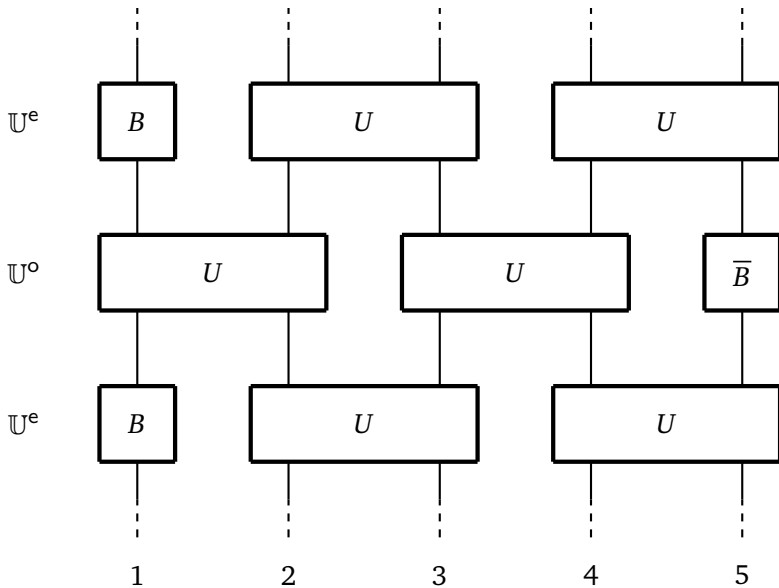

Figure 2: Pictorial representation of the discrete time dynamics for open boundary conditions. Note that the time flows upward.

which amounts to the condition

$$\frac{\alpha_+}{\alpha_-} = \frac{\beta_+}{\beta_-}. \tag{17}$$

In case of the boundary driven system, with boundary matrices corresponding to (13), the distribution (14), (15) is stationary provided that the relation (17) holds. Solution (14), (15) then corresponds to the staggered profile of particle density, which fits the respective boundaries perfectly. For definiteness we shall parametrise $\beta_\pm = \mu\alpha_\pm$, automatically satisfying (17). We also assume normalization $\alpha_+ + \alpha_- + \alpha_0 = 1$, $\beta_+ + \beta_- + \beta_0 = 1$ throughout this section.

Separable states are current carrying states. Indeed, the current of charges, defined as the average number of particles of a given charge crossing a bond, is

$$
\begin{aligned}
J_+ &= \frac{1}{2}\left(\langle +0\rangle_{12} - \langle 0+\rangle_{12}\right) = \frac{1}{2}\alpha_+(1-\mu),\\
J_- &= \frac{1}{2}\left(\langle -0\rangle_{12} - \langle 0-\rangle_{12}\right) = \frac{1}{2}\alpha_-(1-\mu),
\end{aligned}
\tag{18}
$$

where $\langle \tau\tau'\rangle_{12}$ is the joint probability of having a $\tau$ particle at site 1 and $\tau'$ particle at site 2, $\tau, \tau' = 0, +, -$. Note that the factor $1/2$ appears due to the fact that during one unit of time only one half of the bonds are involved in the dynamics.

Below we shall describe several shock types involving the separable states. This will provide an intuitive physical understanding of the exact computations performed in sections 3, 4 and 5.

**Shock between current-carrying separable state and separable state with no current** A separable state satisfying (17), characterized by parameters $\boldsymbol{\alpha} = (\alpha_0, \alpha_+, \alpha_-)$ and $\boldsymbol{\beta} = (\beta_0, \beta_+, \beta_-)$, will be called $\boldsymbol{p}_{[i,j]}(\boldsymbol{\alpha}, \boldsymbol{\beta})$, where $[i, j]$ is the support of the state (*i.e.* the sublattice (interval) of sites on which the state is specified/localized). A special case of a separable states is the state $\boldsymbol{p}_{[i,j]}(\boldsymbol{\alpha}, \boldsymbol{\alpha})$ with $\mu = 1$, *i.e.* $\alpha_\tau = \beta_\tau$, $\tau = 0, \pm$. It corresponds to a constant density profile. A domain wall connecting $\boldsymbol{p}_{[-\infty,i]}(\boldsymbol{\beta}, \boldsymbol{\alpha})$ to $\boldsymbol{p}_{[i+1,\infty]}(\boldsymbol{\beta}, \boldsymbol{\beta})$ has very simple dynamics: it remains sharp and it moves with velocity $v = +1$ or $v = -1$ depending

on whether it is initially located at an odd or even site (due to the staggered nature of the propagator). Indeed, the action of the evolution operator on the respective domain wall is

$$
\begin{aligned}
\mathbb{U}\, p_{[-\infty,0]}(\boldsymbol{\beta},\boldsymbol{\alpha}) \otimes p_{[1,\infty]}(\boldsymbol{\beta},\boldsymbol{\beta}) &= p_{[-\infty,-2]}(\boldsymbol{\beta},\boldsymbol{\alpha}) \otimes p_{[-1,\infty]}(\boldsymbol{\beta},\boldsymbol{\beta}), \\
\mathbb{U}\, p_{[-\infty,-1]}(\boldsymbol{\beta},\boldsymbol{\alpha}) \otimes p_{[0,\infty]}(\boldsymbol{\beta},\boldsymbol{\beta}) &= p_{[-\infty,1]}(\boldsymbol{\beta},\boldsymbol{\alpha}) \otimes p_{[2,\infty]}(\boldsymbol{\beta},\boldsymbol{\beta}).
\end{aligned}
\tag{19}
$$

We note here that in the context of hydrodynamics, such microscopically sharp moving interfaces are named contact discontinuities, and in our model they appear due to the free propagation of vacancies in the (disordered) environment of $+$ and $-$ particles with velocity $v = \pm 1$, i.e. the light-cone velocity. The respective normal modes are thus interaction-free modes, and their appearance can be seen in all quench scenarios considered in this paper, see Figs. 3,4, at the borders of the light cone $x = \pm t$.

Likewise, the domain wall between $p_{[-\infty,i]}(\boldsymbol{\alpha},\boldsymbol{\alpha})$ and $p_{[i+1,\infty]}(\boldsymbol{\beta},\boldsymbol{\alpha})$, is moving with the velocity $\pm 1$, depending on the initial position,

$$
\begin{aligned}
\mathbb{U}\, p_{[-\infty,0]}(\boldsymbol{\alpha},\boldsymbol{\alpha}) \otimes p_{[1,\infty]}(\boldsymbol{\beta},\boldsymbol{\alpha}) &= p_{[-\infty,2]}(\boldsymbol{\alpha},\boldsymbol{\alpha}) \otimes p_{[3,\infty]}(\boldsymbol{\beta},\boldsymbol{\alpha}), \\
\mathbb{U}\, p_{[-\infty,1]}(\boldsymbol{\alpha},\boldsymbol{\alpha}) \otimes p_{[2,\infty]}(\boldsymbol{\beta},\boldsymbol{\alpha}) &= p_{[-\infty,-1]}(\boldsymbol{\alpha},\boldsymbol{\alpha}) \otimes p_{[0,\infty]}(\boldsymbol{\beta},\boldsymbol{\alpha}).
\end{aligned}
\tag{20}
$$

**Shock between two current-carrying separable states**    Let us join two separable states of type $p_{[-\infty,i]}(\boldsymbol{\alpha},\boldsymbol{\gamma})$, with $\gamma_\pm = \mu\alpha_\pm$, on the left and $p_{[i+1,\infty]}(\boldsymbol{\delta},\boldsymbol{\beta})$, with $\delta_\pm = \nu\beta_\pm$, on the right. They carry the currents

$$
\begin{aligned}
J_+^L &= \frac{1}{2}\alpha_+(1-\mu), \\
J_+^R &= -\frac{1}{2}\beta_+(1-\nu),
\end{aligned}
\tag{21}
$$

respectively. Due to the current mismatch, an interface or two separate interfaces must be formed, corresponding to shocks or rarefaction waves, or both (due to existence of two conservation laws for $+$ and $-$ particles). Their motion is regulated by the mass conservation condition. Supposing that the interface will form a single shock between the two current carrying states, we find the interface velocity $v_{sh}^+$ between $+$ particles from the Rankine-Hugoniot condition [23]

$$
v_{sh}^+ = \frac{J_+^R - J_+^L}{\rho_+^R - \rho_+^L} = \frac{\beta_+(\nu-1) - \alpha_+(1-\mu)}{\beta_+(1+\nu) - \alpha_+(1+\mu)}.
\tag{22}
$$

Likewise, the interface velocity $v_{sh}^-$ between $-$ particles is

$$
v_{sh}^- = \frac{J_-^R - J_-^L}{\rho_-^R - \rho_-^L} = \frac{\beta_-(\nu-1) - \alpha_-(1-\mu)}{\beta_-(1+\nu) - \alpha_-(1+\mu)}.
\tag{23}
$$

Note that we used

$$
\begin{aligned}
\rho_\pm^L &= (\alpha_\pm + \mu\alpha_\pm)/2, \\
\rho_\pm^R &= (\beta_\pm + \nu\beta_\pm)/2.
\end{aligned}
\tag{24}
$$

For a single shock scenario to be consistent we must have

$$
v_{sh}^- = v_{sh}^+ = v_{sh},
\tag{25}
$$

giving a consistency condition on the parameters of the separable states. Other consistency conditions are obtained requiring stability of the shock, according to a standard Lax theory of hydrodynamic shocks [23]. It is formulated in terms of characteristic velocities of infinitesimal perturbations on the top of a homogeneous background. The characteristic velocities are eigenvalues of the flux Jacobian $DJ_{\epsilon,\epsilon'} = \partial J_\epsilon / \partial \rho_{\epsilon'}$, for $\epsilon, \epsilon' = \pm$. Using (18), (24), for

the left separable state we find $J_\pm^L = \rho_\pm^L \frac{1-\mu}{1+\mu}$, and consequently the flux Jacobian is diagonal $DJ_{\epsilon,\epsilon'}^L = \frac{1-\mu}{1+\mu}\delta_{\epsilon,\epsilon'}$. The corresponding characteristic velocities are equal $c_1^L = c_2^L = \frac{1-\mu}{1+\mu}$. Similarly, for the right separable state we find $DJ_{\epsilon,\epsilon'}^R = \frac{\nu-1}{1+\nu}\delta_{\epsilon,\epsilon'}$, so that the characteristic velocities are $c_1^R = c_2^R = \frac{\nu-1}{1+\nu}$. The Lax criterium for shock stability requires $\max(c_1^L, c_2^L) \geq v_{sh}$, $\min(c_1^R, c_2^R) \leq v_{sh}$, thus giving

$$\frac{\nu-1}{1+\nu} \leq v_{sh} \leq \frac{1-\mu}{1+\mu}. \tag{26}$$

The condition (26) guarantees that small perturbations generated at both sides of a shock are absorbed by the shock interface, and therefore shock remains stable. Violating (26) results in a smoothening of the interface and to a formation of a rarefaction wave, thus leading to shock instability. The case when one or two characteristic velocities coincide with the shock velocity, is at the border between the two possible scenarios, and will be referred to as marginally stable shock situation.

## 2.3 Out of equilibrium problems: quenches and boundary driven stationary state

In this subsection we present three non-equilibrium problems that we will study analytically. The first two are related to the study of inhomogeneous quench and local quench protocols. The third problem concerns the stationary state of the boundary driven model.

In the case of the *inhomogeneous quench*, we study the situation where at $t = 0$ the system has a density of vacancies $\alpha_0$, the density of positively charged particles $\alpha_+$, and the density of negatively charged particles $\alpha_-$ on the non-positive sites and densities $\beta_0$, $\beta_+$ and $\beta_-$ on positive sites. The parameters satisfy $\alpha_0 + \alpha_+ + \alpha_- = 1$ and $\beta_0 + \beta_+ + \beta_- = 1$. In mathematical terms the initial state (probability vector) is given by

$$\boldsymbol{p}(0) = \bigotimes_{-L/2 < i \leq 0} \begin{pmatrix} \alpha_0 \\ \alpha_+ \\ \alpha_- \end{pmatrix} \otimes \bigotimes_{0 < i \leq L/2} \begin{pmatrix} \beta_0 \\ \beta_+ \\ \beta_- \end{pmatrix}, \tag{27}$$

where the sites labelling should be understood modulo $L$ (we have used negative labelling for later convenience). The initial state is obtained by joining two different stationary measures. Indeed, we have:

$$U \begin{pmatrix} \alpha_0 \\ \alpha_+ \\ \alpha_- \end{pmatrix} \otimes \begin{pmatrix} \alpha_0 \\ \alpha_+ \\ \alpha_- \end{pmatrix} = \begin{pmatrix} \alpha_0 \\ \alpha_+ \\ \alpha_- \end{pmatrix} \otimes \begin{pmatrix} \alpha_0 \\ \alpha_+ \\ \alpha_- \end{pmatrix}, \qquad U \begin{pmatrix} \beta_0 \\ \beta_+ \\ \beta_- \end{pmatrix} \otimes \begin{pmatrix} \beta_0 \\ \beta_+ \\ \beta_- \end{pmatrix} = \begin{pmatrix} \beta_0 \\ \beta_+ \\ \beta_- \end{pmatrix} \otimes \begin{pmatrix} \beta_0 \\ \beta_+ \\ \beta_- \end{pmatrix}, \tag{28}$$

implying that at time $t$ only the sites $-t < i \leq t$ are non-trivially affected by the dynamics (we have a propagation cone) and the state of the system can be written as[1]

$$\boldsymbol{p}(t) = \bigotimes_{i \leq -t} \begin{pmatrix} \alpha_0 \\ \alpha_+ \\ \alpha_- \end{pmatrix} \otimes \boldsymbol{\psi}(t) \otimes \bigotimes_{i > t} \begin{pmatrix} \beta_0 \\ \beta_+ \\ \beta_- \end{pmatrix}. \tag{29}$$

In section 3 we provide an exact expression of the vector $\boldsymbol{\psi}(t) \in \left(\mathbb{C}^3\right)^{\otimes 2t}$. It encodes non-trivial particle currents between the left and the right parts of the system. We use the exact expression to compute analytically the time-dependent density profiles.

---

[1]For the sake of simplicity we omit to describe the second propagation cone emerging from site $L/2$. We recall that we will always consider the case where the time $t < L/4$, *i.e.* when the two propagation cones did not merge yet (the system then behaves as it was on an infinite line).

For the *local quench*, we are interested in the situation where at $t = 0$ the system has a density $\rho_0$ of vacancies, $\rho_+$ of positively charged particles and $\rho_-$ of negatively charged particles on all sites of the lattice except on site 1 where the densities are $\lambda_0$, $\lambda_+$ and $\lambda_-$, with the parameters satisfying the normalization conditions $\rho_0 + \rho_+ + \rho_- = 1$ and $\lambda_0 + \lambda_+ + \lambda_- = 1$. In this case the initial state (probability vector) takes the following form

$$p(0) = \bigotimes_{-L/2 < i \leq 0} \begin{pmatrix} \rho_0 \\ \rho_+ \\ \rho_- \end{pmatrix} \otimes \begin{pmatrix} \lambda_0 \\ \lambda_+ \\ \lambda_- \end{pmatrix} \otimes \bigotimes_{1 < i \leq L/2} \begin{pmatrix} \rho_0 \\ \rho_+ \\ \rho_- \end{pmatrix}. \tag{30}$$

The initial state can be seen as a local defect at site 1 of an equilibrium state (*i.e.* of an invariant measure), since again the stationarity condition

$$U \begin{pmatrix} \rho_0 \\ \rho_+ \\ \rho_- \end{pmatrix} \otimes \begin{pmatrix} \rho_0 \\ \rho_+ \\ \rho_- \end{pmatrix} = \begin{pmatrix} \rho_0 \\ \rho_+ \\ \rho_- \end{pmatrix} \otimes \begin{pmatrix} \rho_0 \\ \rho_+ \\ \rho_- \end{pmatrix} \tag{31}$$

is satisfied. It implies that at time $t$ only the sites $-t < i \leq t$ are non-trivially affected by the dynamics (we have a propagation cone) and the state of the system can be written

$$p(t) = \bigotimes_{i \leq -t} \begin{pmatrix} \rho_0 \\ \rho_+ \\ \rho_- \end{pmatrix} \otimes \psi(t) \otimes \bigotimes_{i > t} \begin{pmatrix} \rho_0 \\ \rho_+ \\ \rho_- \end{pmatrix}. \tag{32}$$

In section 4 we provide an exact expression of the vector $\psi(t) \in \left(\mathbb{C}^3\right)^{\otimes 2t}$, which we then use to calculate the time-dependent density profiles.

The last out-of-equilibrium problem that we will consider is the computation of the stationary state of the model with stochastic boundary conditions that we introduced in the previous subsection. The operator $\mathbb{U} = \mathbb{U}^e \mathbb{U}^o$ indeed defines an irreducible Markov process that admits a unique stationary state $p \in \left(\mathbb{C}^3\right)^L$, *i.e.* a unique probability vector satisfying $\mathbb{U}p = p$. The probability vector $p(t)$ is converging in the long-time limit toward this stationary state for any initial condition $p(0)$. More precisely, we are looking for a pair of vector $p, p'$ satisfying

$$\mathbb{U}^o p = \mu p', \qquad \mathbb{U}^e p' = \frac{1}{\mu} p. \tag{33}$$

This would obviously imply that $p$ is the stationary state. In the section 5 we provide an exact construction of the vectors $p$ and $p'$. It allows us to analytically compute the particle currents and the density profiles in the stationary state.

## 2.4 Matrix product ansatz

Here we introduce the method used to construct the exact solutions to the three different out-of-equilibrium problems defined in the above subsection. The technique is called matrix ansatz and has been widely used in the statistical physics community to compute stationary states of stochastic boundary driven models. Here we will also use it to express exactly the time evolution of the probability distribution in the quench protocols. The idea is to express the components of the vector defined in (29)

$$\psi(t) = \sum_{\tau_i \in \{0,+,-\}} \psi^t_{\tau_{-t+1}, \ldots, \tau_t} e_{\tau_{-t+1}} \otimes e_{\tau_{-t+2}} \otimes \cdots \otimes e_{\tau_t} \tag{34}$$

as a product of matrices (contracted with the row and the column vectors on the left and on the right to get a number)

$$\psi^t_{\tau_{-t+1},\dots,\tau_t} = \langle l|V_{\tau_{-t+1}}W_{\tau_{-t+2}}\dots V_{\tau_{t-1}}W_{\tau_t}|r\rangle. \tag{35}$$

Two triples of matrices $V_\tau$, $W_\tau$, $\tau = 0,\pm$, are acting in an auxiliary space (which is different from the physical space of configurations), $\langle l|$ is a row vector of the auxiliary space and $|r\rangle$ is a column vector of the auxiliary space. We will also use a similar matrix ansatz to construct exactly the solution of the local quench protocol defined in (32) and of the stationary state of the boundary driven model. The matrices $V_\tau$, $W_\tau$ and the boundary vectors $\langle l|$ and $|r\rangle$ have to be carefully chosen in order for this ansatz to correctly encode the probability distribution. In the next sections we will provide an explicit expression of the matrices and of the boundary vectors and we will present an algebraic "cancellation scheme" to prove efficiently that the matrix ansatz is indeed correct.

We will now introduce all of the objects that will be needed to construct the matrix product expressions of the probability distributions studied in this paper. First of all we need to specify the auxiliary space on which the matrices and their building blocks will act and in which the boundary vectors will live. We define the Fock space $\mathcal{F} = \text{span}\{|k\rangle\}_{k\geq 0}$ and the creation and annihilation operators $\mathfrak{a}, \mathfrak{a}^\dagger \in \text{End}(\mathcal{F})$,

$$\mathfrak{a} = \sum_{k=0}^{\infty}|k\rangle\langle k+1|, \qquad \mathfrak{a}^\dagger = \sum_{k=0}^{\infty}|k+1\rangle\langle k|, \tag{36}$$

which satisfy the relation $\mathfrak{a}\mathfrak{a}^\dagger = 1$, where 1 is an identity operator on $\mathcal{F}$. We also introduce the projection operator $\mathfrak{s} = 1 - \mathfrak{a}^\dagger\mathfrak{a} = |0\rangle\langle 0|$, which satisfies the relations $\mathfrak{s}\mathfrak{a}^\dagger = 0$ and $\mathfrak{a}\mathfrak{s} = 0$. It proves useful to introduce the *coherent states*

$$|\underline{x}\rangle = \sum_{k=0}^{\infty}x^k|k\rangle, \qquad \langle\underline{x}| = \sum_{k=0}^{\infty}x^k\langle k|, \tag{37}$$

where $x$ is an arbitrary complex number. They satisfy

$$\begin{aligned}
\mathfrak{a}|\underline{x}\rangle &= x\,|\underline{x}\rangle, & \big(\mathfrak{s} + x(1+\mathfrak{a}^\dagger)\big)|\underline{x}\rangle &= (x+1)|\underline{x}\rangle, \\
\langle\underline{x}|\mathfrak{a}^\dagger &= x\langle\underline{x}|, & \langle\underline{x}|\big(\mathfrak{s} + x(1+\mathfrak{a})\big) &= (x+1)\langle\underline{x}|.
\end{aligned} \tag{38}$$

Let us stress the difference between the state $|k\rangle$, $k \in \mathbb{Z}_+$, an element of the canonical basis (called from now on *canonical state*), and the coherent state $|\underline{k}\rangle$, $k \in \mathbb{C}$. They coincide only for $k = 0$. Note that all of these operators and vectors have already proven to be relevant in the context of exact matrix product expression of out-of-equilibrium stationary distributions in stochastic systems [3,5].

We also introduce a finite dimensional auxiliary vector space $\mathbb{C}^2$, with the canonical basis

$$e_1 = \begin{pmatrix} 1 \\ 0 \end{pmatrix}, \quad \text{and} \quad e_2 = \begin{pmatrix} 0 \\ 1 \end{pmatrix}. \tag{39}$$

Additionally, it proves useful to introduce matrices $A, B \in \text{End}(\mathbb{C}^2)$ obeying $BA = 0$, $A^2 = A$ and $B^2 = B$, which can be represented as the following $2 \times 2$ matrices:

$$A = \begin{pmatrix} 1 & 0 \\ 0 & 0 \end{pmatrix} \quad ; \quad B = \begin{pmatrix} 0 & 1 \\ 0 & 1 \end{pmatrix}. \tag{40}$$

All these algebraic objects will be used extensively in the construction of the exact matrix product states in the following sections.

# 3 Inhomogeneous quench

This section is devoted to the study of the inhomogeneous quench protocol introduced above. We provide an exact construction of the time-dependent probability distribution in a matrix product form and we give an analytical expression of the density profiles. The exact results obtained prove rigorously the existence of both ballistic and diffusive transport.

## 3.1 Time-dependent matrix product ansatz

The vector $\boldsymbol{\psi}(t)$ defined in (29) encodes the non-trivial correlations induced by the inhomogeneous quench inside of the propagation cone. The main result of the present subsection is to provide an exact expression of its components (defined in equation (34)) using a matrix product ansatz

$$\psi^t_{\tau_{-t+1},\dots,\tau_t} = \langle l | V_{\tau_{-t+1}} W_{\tau_{-t+2}} \dots V_{\tau_{t-1}} W_{\tau_t} | r \rangle. \tag{41}$$

The matrices $V_\tau, W_\tau$, $\tau = 0, \pm$, are operators acting on the auxiliary space $\mathbb{C}^2 \otimes \mathcal{F}$, while the boundary vectors $\langle l | \in \mathbb{C}^2 \otimes \mathcal{F}^*$ and $|r\rangle \in \mathbb{C}^2 \otimes \mathcal{F}$ single out an element of contracted matrices which yields correct probability amplitudes $\psi^t_{\tau_{-t+1},\dots,\tau_t}$. For the matrices ($\epsilon = \pm$), we have the following explicit matrix representation:

$$
\begin{aligned}
V_0 &= \beta_0 \, \mathbb{I}_2 \otimes 1 = \beta_0 \begin{pmatrix} 1 & 0 \\ 0 & 1 \end{pmatrix}, \\
V_\epsilon &= \alpha_\epsilon A \otimes \mathfrak{a} + \beta_\epsilon B \otimes 1 = \begin{pmatrix} \alpha_\epsilon \, \mathfrak{a} & \beta_\epsilon \\ 0 & \beta_\epsilon \end{pmatrix}, \\
W_0 &= \alpha_0 \, \mathbb{I}_2 \otimes 1 = \alpha_0 \begin{pmatrix} 1 & 0 \\ 0 & 1 \end{pmatrix}, \\
W_\epsilon &= \alpha_\epsilon A \otimes 1 + \beta_\epsilon B \otimes \mathfrak{a}^\dagger = \begin{pmatrix} \alpha_\epsilon & \beta_\epsilon \, \mathfrak{a}^\dagger \\ 0 & \beta_\epsilon \, \mathfrak{a}^\dagger \end{pmatrix},
\end{aligned}
\tag{42}
$$

and for the boundary vectors we have

$$\langle l | = \mathsf{e}_1^t \otimes \langle 0 | = \big( \langle 0 |, \, 0 \big), \qquad |r\rangle = \mathsf{v} \otimes |0\rangle = \begin{pmatrix} |0\rangle \\ |0\rangle \end{pmatrix}, \tag{43}$$

where we have used the canonical basis vectors $\mathsf{e}_j$, $j = 1, 2$ for $\mathbb{C}^2$ and the $B$-eigenvector $\mathsf{v} = \mathsf{e}_1 + \mathsf{e}_2$.

We now show that the matrix product expression (41) indeed provides the correct time-dependent probability distribution. The proof relies on a simple "telescopic scheme" based on three sets of algrebraic relations. In order to present efficiently those relations and the proof, we have to reformulate the matrix product expression using the tensor space formalism. The vector $\boldsymbol{\psi}(t)$ can be expressed as follows

$$\boldsymbol{\psi}(t) = \langle l | \boldsymbol{V}_{-t+1} \boldsymbol{W}_{-t+2} \cdots \boldsymbol{V}_{t-1} \boldsymbol{W}_t | r \rangle, \tag{44}$$

where $\boldsymbol{V}$ and $\boldsymbol{W}$ are the 3-component vectors encompassing the matrices

$$\boldsymbol{V} = \begin{pmatrix} V_0 \\ V_+ \\ V_- \end{pmatrix}, \qquad \boldsymbol{W} = \begin{pmatrix} W_0 \\ W_+ \\ W_- \end{pmatrix}. \tag{45}$$

The subscripts in (44) denote the components of the tensor space (*i.e.* the sites of the lattice) on which the vectors are located. Note that the components of the expression (44) are equivalent

to (41). The first set of algebraic relations concerns the commutation relations between the matrices and can be concisely written using the tensor space formalism

$$U_{i,i+1}\boldsymbol{W}_i\boldsymbol{V}_{i+1} = \boldsymbol{V}_i\boldsymbol{W}_{i+1}, \tag{46}$$

that is to say in components

$$W_\epsilon V_{\epsilon'} = V_\epsilon W_{\epsilon'}, \quad \epsilon,\epsilon' = \pm, \qquad W_0 V_\tau = V_\tau W_0, \qquad W_\tau V_0 = V_0 W_\tau, \quad \tau = \pm, 0. \tag{47}$$

The other two sets of algebraic relations concern the boundary vectors on the left and on the right and read

$$\langle l|\begin{pmatrix}\alpha_0\\\alpha_+\\\alpha_-\end{pmatrix} = \langle l|\boldsymbol{W}, \qquad \begin{pmatrix}\beta_0\\\beta_+\\\beta_-\end{pmatrix}|r\rangle = \boldsymbol{V}|r\rangle, \tag{48}$$

or equivalently

$$\langle l|\alpha_\tau = \langle l|W_\tau, \qquad \beta_\tau|r\rangle = V_\tau|r\rangle. \tag{49}$$

These relations can be checked by direct computation.

The algebraic relations ensure that the propagation equation is fulfilled

$$U_{-t,-t+1}U_{-t+2,-t+3}\cdots U_{t-2,t-1}U_{t,t+1}\begin{pmatrix}\alpha_0\\\alpha_+\\\alpha_-\end{pmatrix}\otimes\boldsymbol{\psi}(t)\otimes\begin{pmatrix}\beta_0\\\beta_+\\\beta_-\end{pmatrix} = \boldsymbol{\psi}(t+1). \tag{50}$$

The matrix product structure of the time-dependent probability distribution could appear complicated at first sight but it turns out to be very efficient to compute analytically physical quantities such as density profiles.

## 3.2 Exact expression of physical observables

The first step toward the exact computation of physical observables is to verify that the matrix product construction of the above subsection defines a well-normalized probability distribution. The normalization $Z_t$ is defined as the sum of entries of the vector $\boldsymbol{\psi}(t)$. It can be checked from the explicit expression of the matrices (42) and boundary vectors (43) that $\boldsymbol{\psi}(t)$ is correctly normalized, *i.e.*

$$Z_t = \sum_{\tau_{-t+1},\ldots,\tau_t=0,\pm}\psi^t_{\tau_{-t+1},\ldots,\tau_t} = \langle l|T^t|r\rangle = (\alpha_0+\alpha)^t(\beta_0+\beta)^t = 1, \tag{51}$$

where $T = (V_0+V_++V_-)(W_0+W_++W_-)$. In order to lighten the notations, we have introduced the parameters $\alpha = \alpha_+ + \alpha_-$ and $\beta = \beta_+ + \beta_-$, which will be used throughout the paper. The details of the computation are given in appendix A.1.

The probability to observe a particle of species $\epsilon$ at site $j = 2k - t$, with $1 \le k \le t$, can be easily expressed within the matrix product framework as

$$n^{\mathrm{e}}_\epsilon(j,t) = \frac{1}{Z_t}\langle l|T^{k-1}(V_0+V_++V_-)W_\epsilon T^{t-k}|r\rangle. \tag{52}$$

This formula can be exactly evaluated using the explicit expression of the matrices and boundary vectors, see appendix A.2

$$n^{\mathrm{e}}_\epsilon(j,t) = \beta_\epsilon\frac{\alpha}{\beta} + \left(\alpha_\epsilon - \beta_\epsilon\frac{\alpha}{\beta}\right)(1-\beta)^k\alpha^{t-k}\sum_{n=0}^{t-k}\sum_{i=0}^{\min(n,k)}\binom{k}{i}\binom{t-k}{n-i}\left(\frac{1-\alpha}{\alpha}\right)^{n-i}\left(\frac{\beta}{1-\beta}\right)^i. \tag{53}$$

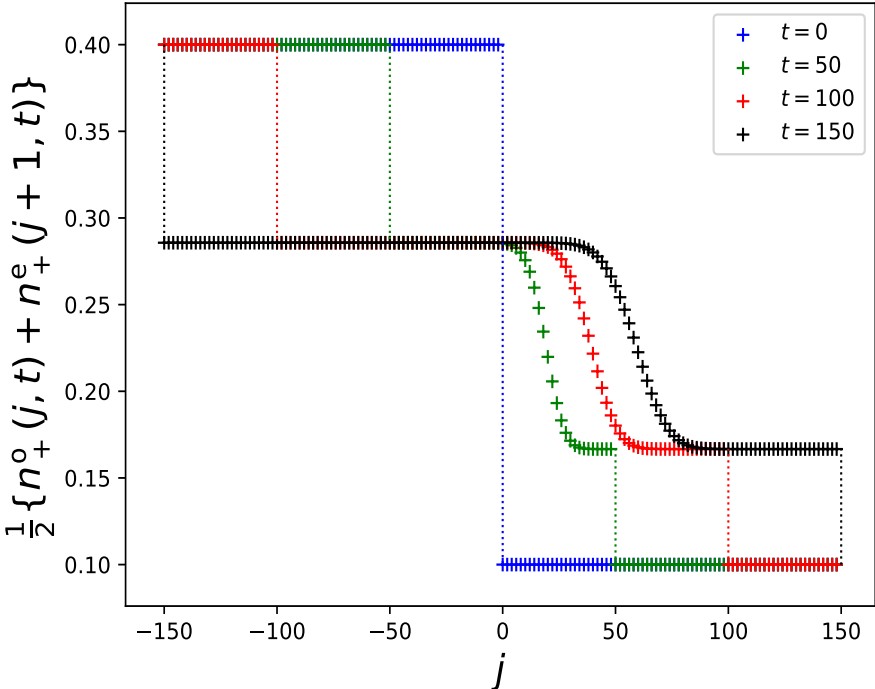

Figure 3: Density profile at different times. The values of the parameters are $\alpha_+ = 0.4$, $\alpha_- = 0.3$, $\beta_+ = 0.1$, $\beta_- = 0.2$. We observe two discontinuities propagating at velocities $-1$ and $1$ respectively (borders of the propagation cone) and a diffusive smoothing of the domain wall on the ray $j = \frac{\alpha-\beta}{\alpha+\beta} t$ inside the propagation cone.

Note that the total particle density is constant

$$n_+^{\mathrm{e}}(j,t) + n_-^{\mathrm{e}}(j,t) = \alpha, \tag{54}$$

and does not depend on time or on the position (as long as $t - j$ is even). This is due to the fact that the total particle density is equal to one minus the density of vacancies. Indeed, the vacancies are propagating at velocity one (they do not interact) and hence their density at any time is trivially deduced from their density at initial time and is equal to $1 - \alpha$ for $t - j$ even.

The probability to observe a particle of species $\epsilon$ at site $j = 2k + 1 - t$, with $0 \leq k \leq t - 1$, can be exactly computed using the matrix product expression, see appendix A.2

$$n_\epsilon^{\mathrm{o}}(j,t) = \beta_\epsilon + \left(\frac{\beta}{\alpha}\alpha_\epsilon - \beta_\epsilon\right)(1-\beta)^k \alpha^{t-k} \sum_{n=0}^{t-k-1} \sum_{i=0}^{\min(n,k)} \binom{k}{i}\binom{t-k}{n-i}\left(\frac{1-\alpha}{\alpha}\right)^{n-i}\left(\frac{\beta}{1-\beta}\right)^i. \tag{55}$$

Again, the total particle density has a very simple expression

$$n_+^{\mathrm{o}}(j,t) + n_-^{\mathrm{o}}(j,t) = \beta, \tag{56}$$

which does not depend on the time and on the position (as long as $t - j$ is odd).

Note that outside the propagation cone, for $j \leq -t$ or $j > t$, it is straightforward to compute

$$n_\epsilon^{\mathrm{e}}(j,t) = n_\epsilon^{\mathrm{o}}(j,t) = \begin{cases} \alpha_\epsilon, & j \leq -t, \\ \beta_\epsilon, & j > t. \end{cases} \tag{57}$$

### 3.3 Hydrodynamic limit

This subsection is devoted to the study of the large time limit of the density profile. For this purpose we will use an equivalent expression of the density, see appendix A.2

$$n^{\text{e}}_{\epsilon}(j,t) = \beta_{\epsilon}\frac{\alpha}{\beta} + \oint \frac{dz}{2i\pi z}\frac{\beta\alpha_{\epsilon} - \alpha\beta_{\epsilon}}{\beta - z\alpha}(1 - \beta + z\alpha)^k\left(\frac{\beta}{z} + 1 - \alpha\right)^{t-k}, \qquad (58)$$

for $j = 2k - t$, i.e. in the case where $t - j$ is even. A similar expression also exists for $j = 2k + 1 - t$

$$n^{\text{o}}_{\epsilon}(j,t) = \beta_{\epsilon} + \oint \frac{dz}{2i\pi}\frac{\beta\alpha_{\epsilon} - \alpha\beta_{\epsilon}}{\beta - z\alpha}(1 - \beta + z\alpha)^k\left(\frac{\beta}{z} + 1 - \alpha\right)^{t-k}. \qquad (59)$$

The asymptotic behavior of the contour integral on the Euler scale $j = xt$, with $-1 \le x \le 1$, and $t$ goes to infinity, can be studied using the saddle point analysis, which yields

$$\lim_{t\to\infty} n^{\text{e}}_{\epsilon}(xt,t) = \begin{cases} \alpha_{\epsilon}, & \text{if } x < \dfrac{\alpha - \beta}{\alpha + \beta}, \\[2mm] \beta_{\epsilon}\dfrac{\alpha}{\beta}, & \text{if } x > \dfrac{\alpha - \beta}{\alpha + \beta}, \end{cases} \qquad (60)$$

$$\lim_{t\to\infty} n^{\text{o}}_{\epsilon}(xt,t) = \begin{cases} \alpha_{\epsilon}\dfrac{\beta}{\alpha}, & \text{if } x < \dfrac{\alpha - \beta}{\alpha + \beta}, \\[2mm] \beta_{\epsilon}, & \text{if } x > \dfrac{\alpha - \beta}{\alpha + \beta}. \end{cases} \qquad (61)$$

At the junction of the two density profiles $x_0 = \frac{\alpha - \beta}{\alpha + \beta}$, where we observe a discontinuity on the Euler scale, the profile exhibits a smooth transition on the diffusive scale $\frac{x - x_0}{t^2}$ [19, 20]. More precisely, for $j = x_0 t + y\sqrt{t}$, we have

$$\lim_{t\to\infty} n^{\text{e}}_{\epsilon}(x_0 t + y\sqrt{t}, t) = \beta_{\epsilon}\frac{\alpha}{\beta} + \frac{1}{2}\left(\alpha_{\epsilon} - \beta_{\epsilon}\frac{\alpha}{\beta}\right)\left(1 - \text{erf}\left(y\sqrt{\frac{(\alpha + \beta)^3}{8\alpha\beta(2 - \alpha - \beta)}}\right)\right), \qquad (62)$$

where we recall the definition of the error function

$$\text{erf}(x) = \frac{2}{\sqrt{\pi}}\int_0^x e^{-u^2}\,du. \qquad (63)$$

Similarly, we have

$$\lim_{t\to\infty} n^{\text{o}}_{\epsilon}(x_0 t + y\sqrt{t}, t) = \beta_{\epsilon} + \frac{1}{2}\left(\alpha_{\epsilon}\frac{\beta}{\alpha} - \beta_{\epsilon}\right)\left(1 - \text{erf}\left(y\sqrt{\frac{(\alpha + \beta)^3}{8\alpha\beta(2 - \alpha - \beta)}}\right)\right). \qquad (64)$$

The equations (62) and (64) are proven starting from the contour integral expressions (58) and (59), using the expansion $(\beta - \alpha z)^{-1} = \beta^{-1}\sum_{l=0}^{\infty}(z\alpha/\beta)^l$. A saddle point analysis reveals that the terms of the sum with non-vanishing contribution in the limit $t \to \infty$ are those for which $l$ is of the order $\sqrt{t}$, and yields the desired formulas.

It is instructive to characterize Fig. 3 from the hydrodynamic point of view, and interpret it in terms of the Riemann problem for a fluid with two locally conserved species (+ and - particles). As an outcome of temporal evolution of initially sharp profile $(\alpha_+, \alpha_-|\beta_+, \beta_-)$ we have: two contact discontinuities of the type Eq.(20), Eq.(19), propagating along the light cone, and a domain wall (discontinuous on the Euler scale) between the two separable current carrying states of the type described in subsec. 2.2. In the following we shall denote a generic

separable state from (14) with $\beta_\pm = \mu\alpha_\pm$ as $(\alpha_\pm; \beta_\pm) \equiv (\alpha_\pm; \mu\alpha_\pm)$, and an interface between two separable states $(\alpha_\pm; \beta_\pm)$ and $(\gamma_\pm; \delta_\pm)$ as $(\alpha_\pm; \beta_\pm | \gamma_\pm; \delta_\pm)$.

With the above notations, on the left boundary of the light cone we have a contact discontinuity between a separable state with no current and a current-carrying separable state $(\alpha_\pm; \alpha_\pm | \frac{\alpha}{\beta}\alpha_\pm; \alpha_\pm)$ of the type (20), propagating to the left with light velocity $-1$, that can be read off from Eqs.(60), (61). Likewise, at the right boundary of the light cone we have a contact discontinuity $(\beta_\pm; \frac{\beta}{\alpha}\beta_\pm | \beta_\pm; \beta_\pm)$, propagating to the right with the velocity of light $+1$. As discussed in subsec. 2.2, these contact discontinuities correspond to interaction-free normal modes due to free propagation of the vacancies. Note that existence of two modes for vacancy propagation is due to even-odd staggered nature of the propagator.

Finally, there is a single diffusive domain wall (discontinuous on Euler scale), of the type $(\frac{\alpha}{\beta}\alpha_\pm; \alpha_\pm | \beta_\pm; \frac{\beta}{\alpha}\beta_\pm)$. Its velocity can be computed from the mass conservation (22), (25) and it gives

$$v_{sh}^- = v_{sh}^+ = \frac{\alpha - \beta}{\alpha + \beta}, \tag{65}$$

in accordance with exact analytic results (60), (61). In addition we find the characteristic velocities of the diffusive modes at both sides of the shock to be $c_1^L = c_1^R = \frac{\alpha-\beta}{\alpha+\beta} = v_{sh}$. According to the Lax criterium, the shock on Fig. 3 is marginally stable (a diffusive discontinuity is parallel to the mode velocities on both sides of the discontinuity).

# 4  Local quench

This section is devoted to the study of the local quench protocol introduced above. Once again we provide an exact construction of the time-dependent probability distribution in a matrix product form and we give an analytical expression of the density profiles.

## 4.1  Time-dependent matrix product ansatz

The vector $\psi(t)$ defined in (32) encodes the non-trivial correlations induced by the local quench in the propagation cone. The main result of the present subsection is to provide an exact expression of its components (defined in equation (34)) using a matrix product ansatz

$$\psi_{\tau_{-t+1},\dots,\tau_t}^t = \langle l|_{\tau_{-t+1}} W_{\tau_{-t+2}} V_{\tau_{-t+3}} \dots V_{\tau_{t-1}} W_{\tau_t} |r\rangle. \tag{66}$$

Note that the ansatz is a bit more complicated than previously because it involves a boundary vector $\langle l|_\tau$ which depends on the content of the site $-t+1$. This dependence is reminiscent of the defect in the initial condition and could be equivalently encoded using a defect matrix (i.e different from $V_\tau, W_\tau$) on the site $-t+1$. The matrices $V_\tau, W_\tau$, $\tau = 0, \pm$, are again operators acting in the auxiliary space $\mathbb{C}^2 \otimes \mathcal{F}$. The boundary vectors $\langle l|_\tau \in \mathbb{C}^2 \otimes \mathcal{F}^*$ and $|r\rangle \in \mathbb{C}^2 \otimes \mathcal{F}$ are used to contract the matrix product ansatz, thus yielding the components of the probability distribution. We choose the following explicit expression for the matrices

$$V_0 = W_0 = \rho_0 \mathsf{P} \quad \text{where} \quad \mathsf{P} = \begin{pmatrix} 1-\mathfrak{s} & 0 \\ 0 & 1 \end{pmatrix},$$

$$V_\pm = \mathsf{P}\begin{pmatrix} \rho_\pm\mathfrak{a} & \lambda_\pm \\ 0 & \rho_\pm \end{pmatrix} \quad \text{and} \quad W_\pm = \mathsf{P}\begin{pmatrix} \rho_\pm & \lambda_\pm\mathfrak{a}^\dagger \\ 0 & \rho_\pm\mathfrak{a}^\dagger \end{pmatrix}. \tag{67}$$

Note that $\mathsf{P}$ is a projector ($\mathsf{P}^2 = \mathsf{P}$) obeying $V_\tau \mathsf{P} = V_\tau$ and $W_\tau \mathsf{P} = W_\tau$, $\tau = \pm, 0$. For the boundary vectors, we choose the following form

$$\langle l|_0 = \begin{pmatrix} 0, \lambda_0\langle\underline{1}| \end{pmatrix}, \qquad \langle l|_\pm = \begin{pmatrix} \rho_\pm\langle 1|, \lambda_\pm\langle 0| \end{pmatrix}, \qquad |r\rangle = \begin{pmatrix} 0 \\ |0\rangle \end{pmatrix}, \tag{68}$$

where we have used a coherent state in $\langle l|_0$, and canonical states in $\langle l|_\pm$ and $|r\rangle$. Note that $\langle l|_\tau P = \langle l|_\tau$, $\tau = \pm, 0$.

In order to efficiently show that the matrix product expression (66) provides the correct time-dependent probability distribution, we reformulate the matrix product expression using the tensor space formalism

$$\psi(t) = \langle l|_{-t+1} W_{-t+2} \cdots V_{t-1} W_t |r\rangle, \tag{69}$$

where $V$, $W$ and $\langle l|$ are the 3-component vectors

$$V = \begin{pmatrix} V_0 \\ V_+ \\ V_- \end{pmatrix}, \qquad W = \begin{pmatrix} W_0 \\ W_+ \\ W_- \end{pmatrix}, \qquad \langle l| = \begin{pmatrix} \langle l|_0 \\ \langle l|_+ \\ \langle l|_- \end{pmatrix}. \tag{70}$$

The subscripts in (69) denote the components of the tensor space (*i.e.* the sites of the lattice) on which the vectors are located. The matrices obey the bulk relations (46). On the left boundary, the following condition should be satisfied

$$\langle l|_{-t} W_{-t+1} = U_{-t,-t+1} \begin{pmatrix} \rho_0 \\ \rho_+ \\ \rho_- \end{pmatrix}_{-t} \langle l|_{-t+1}, \tag{71}$$

that is to say

$$\langle l|_\epsilon W_{\epsilon'} = \rho_\epsilon \langle l|_{\epsilon'}, \quad \epsilon, \epsilon' = \pm, \qquad \langle l|_0 W_\tau = \rho_\tau \langle l|_0, \qquad \langle l|_\tau W_0 = \rho_0 \langle l|_\tau, \quad \tau = \pm, 0. \tag{72}$$

On the right boundary, we have

$$\begin{pmatrix} \rho_0 \\ \rho_+ \\ \rho_- \end{pmatrix} |r\rangle = V |r\rangle \quad i.e. \quad V_\tau |r\rangle = \rho_\tau |r\rangle, \tag{73}$$

and finally, the boundary vectors should also satisfy the 'initial condition'

$$\langle l|_\tau \cdot |r\rangle = \lambda_\tau. \tag{74}$$

The algebraic relations ensure that the propagation equation is fulfilled

$$U_{-t,-t+1} U_{-t+2,-t+3} \cdots U_{t-2,t-1} U_{t,t+1} \begin{pmatrix} \rho_0 \\ \rho_+ \\ \rho_- \end{pmatrix} \otimes \psi(t) \otimes \begin{pmatrix} \rho_0 \\ \rho_+ \\ \rho_- \end{pmatrix} = \psi(t+1). \tag{75}$$

Once again the matrix product structure will prove to be very efficient to analytically compute physical quantities, such as the density profiles.

## 4.2 Exact expression of physical observables

The first step toward the exact computation of the time-dependent density profiles is to check that the vector $\psi(t)$, defining the probability distribution in the propagation cone, is correctly normalized. The normalization $Z_t$ of the vector $\psi(t)$ is defined as the sum of its components and can be written using the matrix product formalism as follows

$$Z_t = \sum_{\tau_{-t+1},\dots,\tau_t = 0,\pm} \psi^t_{\tau_{-t+1},\dots,\tau_t} = (\langle l|_0 + \langle l|_+ + \langle l|_-) T^{t-1} (W_0 + W_+ + W_-)|r\rangle, \tag{76}$$

where $T = (W_0 + W_+ + W_-)(V_0 + V_+ + V_-)$. It can be evaluated, see appendix B.1, using the explicit form of the matrices and of the boundary vectors

$$Z_t = (\rho_0 + \rho)^{2t-1}(\lambda_0 + \lambda) = 1, \tag{77}$$

where we have introduced

$$\rho = \rho_+ + \rho_-, \qquad \lambda = \lambda_+ + \lambda_-, \tag{78}$$

in order to lighten the notations. Those parameters will also be used in the rest of the section. The probability to observe a particle of species $\epsilon$ at site $j = 2k - t$, with $1 \le k \le t$, can be easily expressed within the matrix product framework as

$$n_\epsilon^{\text{e}}(j,t) = \frac{1}{Z_t} (\langle l|_0 + \langle l|_+ + \langle l|_-) T^{k-1} W_\epsilon T^{t-k} |r\rangle. \tag{79}$$

This formula can be exactly evaluated using the explicit expression of the matrices and boundary vectors, see appendix B.2

$$n_\epsilon^{\text{e}}(j,t) = \rho_\epsilon + (\rho\lambda_\epsilon - \lambda\rho_\epsilon) \sum_{l=0}^{\min(k-1,t-k)} \binom{k-1}{l}\binom{t-k}{l} \rho^{2l}(1-\rho)^{t-1-2l}. \tag{80}$$

The total particle density takes the very simple expression

$$n_+^{\text{e}}(j,t) + n_-^{\text{e}}(j,t) = \rho. \tag{81}$$

The probability to observe a particle of species $\epsilon$ at site $j = 2k+1-t$, with $1 \le k \le t-1$, can be exactly computed using the matrix product expression, see appendix B.2

$$n_\epsilon^{\text{o}}(j,t) = \rho_\epsilon + (\rho\lambda_\epsilon - \lambda\rho_\epsilon) \sum_{l=0}^{\min(k-1,t-k-1)} \binom{k-1}{l}\binom{t-k}{l+1} \rho^{2l+1}(1-\rho)^{t-2-2l}. \tag{82}$$

Once again, the total particle density is simply

$$n_+^{\text{o}}(j,t) + n_-^{\text{o}}(j,t) = \rho. \tag{83}$$

The density on the site $j = -t+1$ (i.e. for $k = 0$) takes a specific expression, see appendix B.2

$$n_\epsilon^{\text{o}}(-t+1,t) = \frac{\rho_\epsilon\lambda}{\rho} + \frac{(1-\rho)^t(\rho\lambda_\epsilon - \lambda\rho_\epsilon)}{\rho}. \tag{84}$$

In this case the total particle density reads

$$n_+^{\text{o}}(-t+1,t) + n_-^{\text{o}}(-t+1,t) = \lambda. \tag{85}$$

## 4.3 Hydrodynamic limit

An equivalent alternative expression for the density is given by

$$n_\epsilon^{\text{e}}(j,t) = \rho_\epsilon + (\rho\lambda_\epsilon - \lambda\rho_\epsilon) \oint \frac{dz}{2i\pi z}(1-\rho+\rho z)^{k-1}\left(1-\rho+\frac{\rho}{z}\right)^{t-k}, \tag{86}$$

$$n_\epsilon^{\text{o}}(j,t) = \rho_\epsilon + (\rho\lambda_\epsilon - \lambda\rho_\epsilon) \oint \frac{dz}{2i\pi}(1-\rho+\rho z)^{k-1}\left(1-\rho+\frac{\rho}{z}\right)^{t-k}. \tag{87}$$

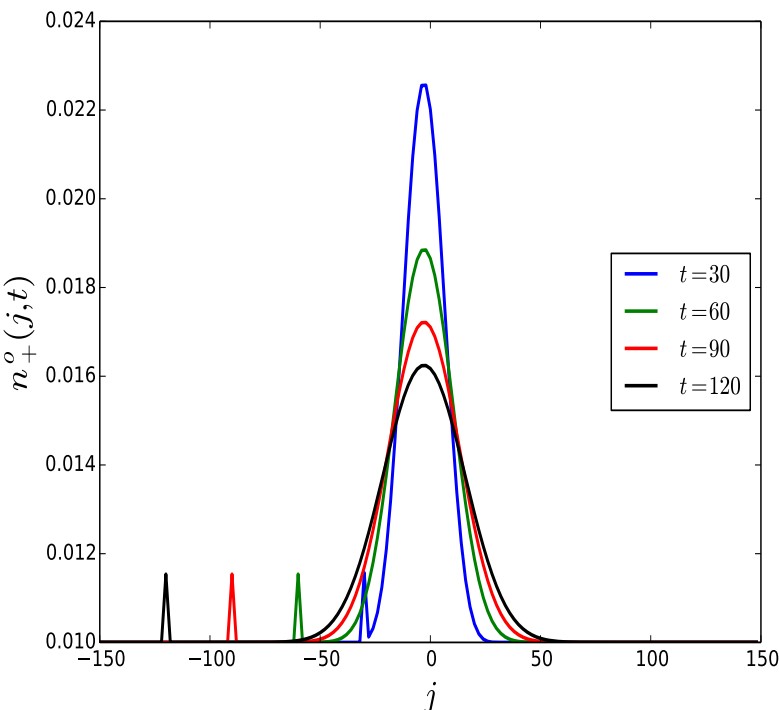

Figure 4: Time evolution of the density profile. The values of the parameters are $\rho_+ = 0.01$, $\rho_- = 0.25$, $\lambda_+ = 0.3$, $\lambda_- = 0$.

Using the saddle point analysis we can again study the asymptotic behavior of the contour integral in the regime where $j = xt$, with $-1 \leq x \leq 1$, and $t$ goes to infinity. The contribution of the contour integral is exponentially small when $x \neq 0$, and one gets

$$\lim_{t \to \infty} n^{\mathrm{e}}_{\epsilon}(xt, t) = \lim_{t \to \infty} n^{\mathrm{o}}_{\epsilon}(xt, t) = \rho_{\epsilon}. \tag{88}$$

Note that at the site $j = -t + 1$, we can obtain an explicit expression, which in the long time limit reads

$$\lim_{t \to \infty} n^{\mathrm{o}}_{\epsilon}(-t+1, t) = \frac{\rho_{\epsilon} \lambda}{\rho}. \tag{89}$$

This corresponds to the asymptotic value of a ballistically propagating peak, which can be observed in Fig. 4. In the vicinity of $x = 0$, in the regime $x = \frac{y}{\sqrt{t}}$, we obtain

$$n^{\mathrm{e}}_{\epsilon}(y\sqrt{t}, t) = \rho_{\epsilon} + (\rho \lambda_{\epsilon} - \lambda \rho_{\epsilon}) \frac{\exp\left(-\frac{\rho y^2}{2(1-\rho)}\right)}{\sqrt{2\pi \rho(1-\rho)t}} + \mathcal{O}(1/t). \tag{90}$$

The asymptotic behavior of $n^{\mathrm{o}}_{\epsilon}(y\sqrt{t}, t)$ is exactly the same.

The ballistic left mover on Fig. 4 on the extremity of the propagation cone can be interpreted as a consequence of the ballistically propagating vacancies. At time $t = 0$ there is a discrepancy between the density at site 1 and the density elsewhere. The latter is propagating ballistically to the left because of the staggered nature of the dynamics (if the local defect was initially located on an even site the ballistic peak would be moving to the right).

# 5 Stochastic boundary driving

## 5.1 Matrix product expression of the stationary state

We recall that we are using the short-hand notations

$$\alpha = \alpha_+ + \alpha_- \quad \text{and} \quad \beta = \beta_+ + \beta_-. \tag{91}$$

The main result of the present subsection is an exact expression of the components of the stationary states $p$, defined by the equations $\mathbb{U}p = p$ (where the operator $\mathbb{U}$ has been defined in the subsection 2.3). We have

$$p_{\tau_1,\tau_2,\ldots,\tau_L} = \langle l|_{\tau_1} V_{\tau_2} W_{\tau_3} V_{\tau_4} \cdots V_{\tau_{L-3}} W_{\tau_{L-2}} |rr\rangle_{\tau_{L-1},\tau_L}, \tag{92}$$

where the matrices $V_\tau, W_\tau \in \text{End}(\mathbb{C}^2 \otimes \mathcal{F} \otimes \mathcal{F})$. Note that the auxiliary space is now a bit more complicated than in the previous cases because it involves two copies of the infinite dimensional Fock space $\mathcal{F}$. The matrix product ansatz is constructed by employing matrices $A$ and $B$, which were introduced in the preceding section (40):

$$
\begin{aligned}
V_0 &= \alpha\beta_0 \Big[ \mathbb{I}_2 \otimes 1 \otimes (1 + \mathfrak{a}^\dagger) + A \otimes \mathfrak{a} \otimes \mathfrak{s} \Big], \\
V_\pm &= \alpha_\pm \beta\, A \otimes \mathfrak{a} \otimes (1 + \mathfrak{a}) + \alpha\beta_\pm\, B \otimes 1 \otimes (1 + \mathfrak{a}^\dagger), \\
W_0 &= \alpha_0 \beta \Big[ \mathbb{I}_2 \otimes (1 + \mathfrak{a}) \otimes 1 + B \otimes \mathfrak{s} \otimes \mathfrak{a}^\dagger \Big], \\
W_\pm &= \alpha_\pm \beta\, A \otimes (1 + \mathfrak{a}) \otimes 1 + \alpha\beta_\pm\, B \otimes (1 + \mathfrak{a}^\dagger) \otimes \mathfrak{a}^\dagger.
\end{aligned}
\tag{93}
$$

The vectors $\langle l|_\tau \in \mathbb{C}^2 \otimes \mathcal{F}^* \otimes \mathcal{F}^*$ read explicitly

$$\langle l|_\tau = \alpha_\tau\, \mathsf{e}_1^t \otimes \langle 0| \otimes \underline{\langle \beta/\beta_0|}, \quad \tau = 0, \pm. \tag{94}$$

The vectors $|rr\rangle_{\tau,\tau'} \in \mathbb{C}^2 \otimes \mathcal{F} \otimes \mathcal{F}$ are expressed using the $B$-eigenvector $\mathsf{v} = \mathsf{e}_1 + \mathsf{e}_2$

$$
\begin{aligned}
|rr\rangle_{\tau 0} &= \alpha_0 \beta_\tau \beta\, \mathsf{v} \otimes \Big[ \mathfrak{s} |\underline{\alpha/\alpha_0}\rangle \otimes \mathfrak{a}^\dagger |0\rangle + \Big(1 + \frac{\alpha}{\alpha_0}\Big) |\underline{\alpha/\alpha_0}\rangle \otimes |0\rangle \Big], \quad \tau = 0, \pm, \\
|rr\rangle_{0\pm} &= \alpha\beta_0 \beta_\pm\, \mathsf{v} \otimes (\mathbb{1} + \mathfrak{a}^\dagger) |\underline{\alpha/\alpha_0}\rangle \otimes \mathfrak{a}^\dagger |0\rangle + \beta\beta_0 \alpha_\pm \Big(1 + \frac{\alpha}{\alpha_0}\Big) \mathsf{e}_1 \otimes |\underline{\alpha/\alpha_0}\rangle \otimes |0\rangle, \\
|rr\rangle_{\epsilon\epsilon'} &= \alpha\beta_\epsilon \beta_{\epsilon'}\, \mathsf{v} \otimes (\mathbb{1} + \mathfrak{a}^\dagger) |\underline{\alpha/\alpha_0}\rangle \otimes \mathfrak{a}^\dagger |0\rangle + \beta\beta_{\epsilon'} \alpha_\epsilon \Big(1 + \frac{\alpha}{\alpha_0}\Big) \mathsf{e}_1 \otimes |\underline{\alpha/\alpha_0}\rangle \otimes |0\rangle, \quad \epsilon, \epsilon' = \pm.
\end{aligned}
\tag{95}
$$

In order to prove the matrix product expression of the stationary state it will be useful to reformulate it using the tensor space formalism

$$p = \langle l|_1 V_2 W_3 V_4 \cdots V_{L-3} W_{L-2} |rr\rangle_{L-1,L}, \tag{96}$$

where

$$
\langle l| = \begin{pmatrix} \langle l|_0 \\ \langle l|_+ \\ \langle l|_- \end{pmatrix}, \qquad
V = \begin{pmatrix} V_0 \\ V_+ \\ V_- \end{pmatrix}, \qquad
W = \begin{pmatrix} W_0 \\ W_+ \\ W_- \end{pmatrix}, \qquad
|rr\rangle = \begin{pmatrix} |rr\rangle_{00} \\ |rr\rangle_{0+} \\ |rr\rangle_{0-} \\ |rr\rangle_{+0} \\ |rr\rangle_{++} \\ |rr\rangle_{+-} \\ |rr\rangle_{-0} \\ |rr\rangle_{-+} \\ |rr\rangle_{--} \end{pmatrix}. \tag{97}
$$

We recall that the subscripts indicate on which tensor space components (*i.e.* on which sites of the lattice) the operators are acting. As already mentioned, the proof relies on the construction of a pair of vectors $\boldsymbol{p}$ and $\boldsymbol{p}'$ satisfying the following relations

$$\mathbb{U}^e \boldsymbol{p} = \frac{\beta}{\alpha}(\alpha + \alpha_0)\boldsymbol{p}' \quad \text{and} \quad \mathbb{U}^o \boldsymbol{p}' = \frac{\alpha}{\beta}(\beta + \beta_0)\boldsymbol{p}. \tag{98}$$

These relations imply the following eigenvector relation

$$\mathbb{U}\boldsymbol{p} = (\alpha + \alpha_0)(\beta + \beta_0)\boldsymbol{p}. \tag{99}$$

We already provided the expression of the stationary state $\boldsymbol{p}$. We now give the expression of the vector $\boldsymbol{p}'$. For this purpose we need to introduce the boundary vectors

$$\langle ll| = \begin{pmatrix} \langle ll|_{00} \\ \langle ll|_{0+} \\ \langle ll|_{0-} \\ \langle ll|_{+0} \\ \langle ll|_{++} \\ \langle ll|_{+-} \\ \langle ll|_{-0} \\ \langle ll|_{-+} \\ \langle ll|_{--} \end{pmatrix}, \qquad |r\rangle = \begin{pmatrix} |r\rangle_0 \\ |r\rangle_+ \\ |r\rangle_- \end{pmatrix}, \tag{100}$$

where

$$|r\rangle_\tau = \beta_\tau \mathsf{v} \otimes |\underline{\alpha/\alpha_0}\rangle \otimes |0\rangle, \quad \tau = 0, \pm, \tag{101}$$

and

$$\langle ll|_{0\tau} = \beta_0 \alpha_\tau \alpha \, \mathsf{e}_1^t \otimes \left[ \langle 0|\mathfrak{a} \otimes \langle \underline{\beta/\beta_0}|\mathfrak{s} + \left(1 + \frac{\beta}{\beta_0}\right)\langle 0| \otimes \langle \underline{\beta/\beta_0}| \right],$$

$$\langle ll|_{\pm 0} = \beta \alpha_0 \alpha_\pm \, \mathsf{e}_1^t \otimes \langle 0|\mathfrak{a} \otimes \langle \underline{\beta/\beta_0}|(\mathbb{1} + \mathfrak{a}) + \alpha \alpha_0 \beta_\pm \left(1 + \frac{\beta}{\beta_0}\right) \mathsf{e}_2^t \otimes \langle 0| \otimes \langle \underline{\beta/\beta_0}|, \tag{102}$$

$$\langle ll|_{\epsilon\epsilon'} = \beta \alpha_\epsilon \alpha_{\epsilon'} \, \mathsf{e}_1^t \otimes \langle 0|\mathfrak{a} \otimes \langle \underline{\beta/\beta_0}|(\mathbb{1} + \mathfrak{a}) + \alpha \alpha_\epsilon \beta_{\epsilon'} \left(1 + \frac{\beta}{\beta_0}\right) \mathsf{e}_2^t \otimes \langle 0| \otimes \langle \underline{\beta/\beta_0}|.$$

Building on those, we define

$$\boldsymbol{p}' = \langle ll|_{12} V_3 W_4 V_5 \cdots V_{L-2} W_{L-1} |r\rangle_L. \tag{103}$$

The relations (98) are proven using local exchange relations in the bulk

$$U_{i,i+1} W_i V_{i+1} = V_i W_{i+1}, \tag{104}$$

on the right boundary

$$\overline{B}_L |rr\rangle_{L-1,L} = \frac{\beta}{\alpha}(\alpha + \alpha_0) V_{L-1} |r\rangle_L,$$

$$U_{L-1,L} W_{L-1} |r\rangle_L = |r\rangle_{L-1,L}, \tag{105}$$

and on the left boundary

$$B_1 \langle ll|_{12} = \frac{\alpha}{\beta}(\beta + \beta_0)\langle l|_1 W_2,$$

$$U_{1,2} \langle l|_1 V_2 = \langle ll|_{12}. \tag{106}$$

By direct computation it is possible to check that all of the above relations are satisfied by the ansatz (93), (94) and (95).

## 5.2 Exact expression of physical observables

In order to compute physical quantities we need to evaluate the normalization of the matrix product probability distribution, which corresponds to the sum over all the entries (*i.e* over all configurations) of the stationary matrix product state

$$Z_L = \sum_{\tau_1,\ldots,\tau_L=0,\pm} p_{\tau_1,\ldots,\tau_L} = \left( \sum_{\tau=0,\pm} \langle l|_\tau \right) T^{\frac{L-3}{2}} \left( \sum_{\tau,\tau'=0,\pm} |r\rangle_{\tau,\tau'} \right), \tag{107}$$

where $T = (V_0 + V_+ + V_-)(W_0 + W_+ + W_-)$. We show in appendix C.1 that it is given by the following exact formula

$$Z_L = \frac{\alpha(1-\beta)^{L-1} - \beta(1-\alpha)^{L-1}}{\alpha - \beta} \frac{\alpha^{\frac{L-3}{2}} \beta^{\frac{L-1}{2}}}{(1-\alpha)^{L-2}(1-\beta)^{L-2}}. \tag{108}$$

The mean particle current associated to the species $\epsilon = \pm$ is defined by the mean number of particles of species $\epsilon$ that jump between sites $i$ and $i+1$ during a unit of time. Since the bulk dynamics conserves the particle number, the steady state current does not depend on the site index. We can, for instance, compute it between site 1 and site 2. Using the matrix product expression of the stationary state, the current is given by

$$J_\epsilon = \frac{1}{Z_L} (\langle l|_\epsilon V_0 - \langle l|_0 V_\epsilon)(W_0 + W_+ + W_-) T^{\frac{L-5}{2}} \left( \sum_{\tau,\tau'=0,\pm} |r\rangle_{\tau,\tau'} \right). \tag{109}$$

The above expression can be evaluated exactly (see appendix C.2)

$$J_\epsilon = \frac{\alpha_\epsilon(1-\beta)^{L-1} - \beta_\epsilon(1-\alpha)^{L-1}}{\alpha(1-\beta)^{L-1} - \beta(1-\alpha)^{L-1}} (\alpha - \beta). \tag{110}$$

From this result we can deduce that the total particle current takes a very simple form

$$J_+ + J_- = \alpha - \beta. \tag{111}$$

When $\alpha = \beta$, the current is given by the appropriate limit of the general expression

$$J_\epsilon = \frac{1-\alpha}{1+(L-2)\alpha} (\alpha_\epsilon - \beta_\epsilon). \tag{112}$$

The mean particle density associated to the species $\epsilon = \pm$ at site $i$ is defined by the probability to observe a particle $\epsilon$ at site $i$ in the stationary state. Since we are interested in mean behavior, we need to average the density over two time steps (odd and even): the stationary state is described by $p$ and $p'$ respectively. The density at site $i = 2k + 1$ is thus given in the matrix product formalism by

$$n_\epsilon(i) = \frac{1}{2Z_L} \left( \sum_{\tau\in\{0,+,-\}} \langle l|_\tau \right) T^{k-1}(V_0 + V_+ + V_-)W_\epsilon T^{\frac{L-3}{2}-k} \left( \sum_{\tau,\tau'\in\{0,+,-\}} |rr\rangle_{\tau\tau'} \right)$$
$$+ \frac{1}{2\tilde{Z}_L} \left( \sum_{\tau,\tau'\in\{0,+,-\}} \langle ll|_{\tau\tau'} \right) T^{k-1}V_\epsilon(W_0 + W_+ + W_-)T^{\frac{L-3}{2}-k} \left( \sum_{\tau\in\{0,+,-\}} |r\rangle_\tau \right), \tag{113}$$

where $\tilde{Z}_L$ is the normalization (*i.e.* the sum of the components) of $p'$. Once again the matrix product expression can be computed explicitly (see appendix C.3)

$$n_\epsilon(i) = \frac{1}{2}(\alpha+\beta)\frac{\alpha_\epsilon(1-\beta)^{L-1} - \beta_\epsilon(1-\alpha)^{L-1}}{\alpha(1-\beta)^{L-1} - \beta(1-\alpha)^{L-1}} + \frac{(\alpha\beta_\epsilon - \beta\alpha_\epsilon)(1-\alpha)^{L-i}(1-\beta)^{i-1}}{\alpha(1-\beta)^{L-1} - \beta(1-\alpha)^{L-1}}. \tag{114}$$

We deduce that the total particle density takes a very simple form

$$n_+(i) + n_-(i) = \frac{1}{2}(\alpha + \beta). \tag{115}$$

Note that we have an equivalent expression of the particle density (114)

$$n_\epsilon(i) = \alpha_\epsilon (1-\beta)^{i-1} \frac{\alpha(1-\beta)^{L-i} - \beta(1-\alpha)^{L-i}}{\alpha(1-\beta)^{L-1} - \beta(1-\alpha)^{L-1}} + \beta_\epsilon (1-\alpha)^{L-i} \frac{\alpha(1-\beta)^{i-1} - \beta(1-\alpha)^{i-1}}{\alpha(1-\beta)^{L-1} - \beta(1-\alpha)^{L-1}}$$
$$- \frac{1}{2} \frac{(\alpha - \beta)\big(\alpha_\epsilon (1-\beta)^{L-1} + \beta_\epsilon (1-\alpha)^{L-1}\big)}{\alpha(1-\beta)^{L-1} - \beta(1-\alpha)^{L-1}}, \tag{116}$$

which allows us to easily take the limit $\beta \to \alpha$, and obtain an expression of the particle density in the case $\alpha = \beta$

$$n_\epsilon(i) = \frac{\alpha_\epsilon \left[\frac{1}{2} + \alpha\left(L - i - \frac{1}{2}\right)\right] + \beta_\epsilon \left[\frac{1}{2} + \alpha\left(i - \frac{3}{2}\right)\right]}{1 + \alpha(L-2)}. \tag{117}$$

Note that in this case the density profile is a linear profile interpolating between the densities of the left and right reservoirs $\alpha_\epsilon$ and $\beta_\epsilon$.

## 5.3 Hydrodynamic limit and phase transitions

In this subsection we investigate the properties of the physical quantities in the large system size limit $L \to \infty$. Depending on the boundary parameters, we encounter different scaling behaviors of the particle current with respect to the system size, which are interpreted as signature of ballistic and diffusive transport. More precisely the parameters space splits into three different phases:

**The right reservoir phase, for $\alpha < \beta$.** The particle current has a finite large system size limit

$$\lim_{L \to \infty} J_\epsilon = \frac{\beta_\epsilon}{\beta}(\alpha - \beta). \tag{118}$$

The limit of the particle density is

$$\lim_{L \to \infty} n_\epsilon(i) = \frac{1}{2}\frac{\beta_\epsilon}{\beta}(\alpha + \beta) - \frac{(\alpha\beta_\epsilon - \beta\alpha_\epsilon)}{\beta}\left(\frac{1-\beta}{1-\alpha}\right)^{i-1}. \tag{119}$$

It has a constant value $\frac{1}{2}\frac{\beta_\epsilon}{\beta}(\alpha + \beta)$, fixed by the right reservoir in the bulk, and an exponential tail near the left boundary.

**The left reservoir phase, for $\beta < \alpha$.** The particle current has a finite large system size limit

$$\lim_{L \to \infty} J_\epsilon = \frac{\alpha_\epsilon}{\alpha}(\alpha - \beta). \tag{120}$$

The limit of the particle density is

$$\lim_{L \to \infty} n_\epsilon(L-i) = \frac{1}{2}\frac{\alpha_\epsilon}{\alpha}(\alpha + \beta) + \frac{(\alpha\beta_\epsilon - \beta\alpha_\epsilon)}{\alpha}\left(\frac{1-\alpha}{1-\beta}\right)^{i}. \tag{121}$$

It has a constant value $\frac{1}{2}\frac{\alpha_\epsilon}{\alpha}(\alpha + \beta)$, fixed by the left reservoir, and an exponential tail near the right boundary.

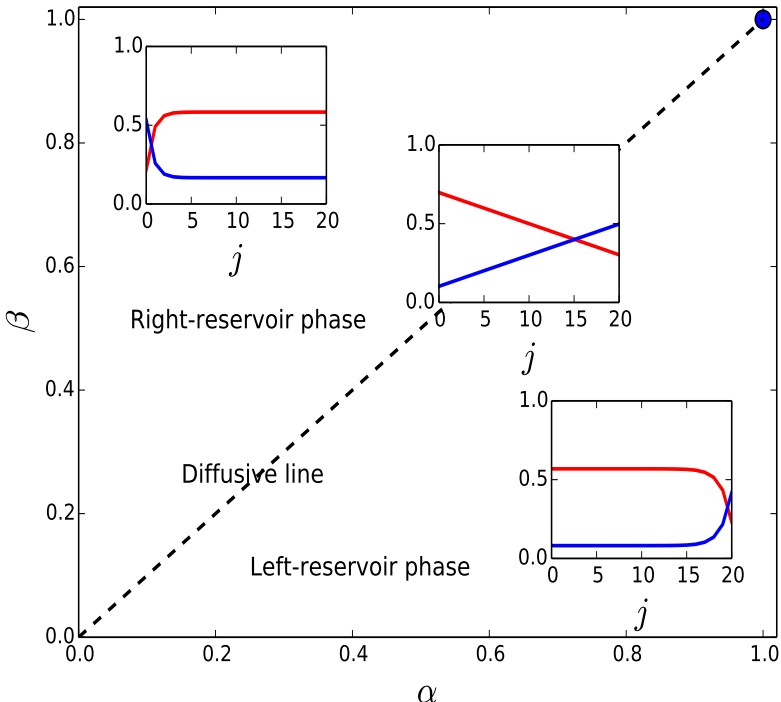

Figure 5: Phase diagram of the boundary driven cellular automaton. For the density plots in the insets we took a lattice of length $L = 21$, and depict density of $+/-$ with red/blue curves The values of the parameters are $\alpha_+ = 0.7$, $\alpha_- = 0.1$, $\beta_+ = 0.1$, $\beta_- = 0.4$ for the density plot in the left reservoir phase, $\alpha_+ = 0.1$, $\alpha_- = 0.5$, $\beta_+ = 0.7$, $\beta_- = 0.2$ for the plot in the right reservoir phase and $\alpha_+ = 0.7$, $\alpha_- = 0.1$, $\beta_+ = 0.3$, $\beta_- = 0.5$ for the plot on the diffusive line. The blue dot on the upper right of the phase diagram represents an insulating point (no vacancies on the lattice).

**The diffusive phase, for $\alpha = \beta$.** In this phase the particle current decays as $1/L$ in the large system size limit. More precisely we have

$$j = \lim_{L \to \infty} L \times J_\epsilon = \frac{1-\alpha}{\alpha}(\alpha_\epsilon - \beta_\epsilon). \tag{122}$$

The limit of the particle density is

$$n(x) = \lim_{L \to \infty} n_\epsilon(Lx) = \alpha_\epsilon(1-x) + \beta_\epsilon x. \tag{123}$$

In this phase the Fick law (which is a signature of the diffusive behavior) is satisfied

$$j = -\mathcal{D}n'(x), \tag{124}$$

where $\mathcal{D} = \frac{1-\alpha}{\alpha}$ is the diffusion constant, in accordance with the value computed in [19, 20].

# 6 Discussion and conclusion

In the article we presented a unified algebraic framework for obtaining exact solutions of different out-of-equilibrium setups in terms of the (time-dependent) matrix product ansatz. In particular, using the new framework we obtained an explicit solution of the hardcore interacting deterministic lattice gas, and calculated physically interesting quantities such as the current

and the density profiles. Interestingly, in the case of the boundary driving we observed that, depending on the parameters of external driving, the system undergoes a phase transition.

Our framework paves a new way for obtaining exact time-dependent results in statistical systems (in particular, with deterministic bulk dynamics). We hope that the approach presented here can be generalized to more complicated systems. The first step on this path would be to understand the time-dependent solution of the classical and quantum Rule 54 automaton [21, 22].

The full time dependent result also enables the calculation of physical quantities beyond the expectation values of currents or charges. In particular it would be interesting to obtain a large deviation functional, especially in the case of the boundary driving, where we observed the phase transition.

Although in this paper we focused on a model with deterministic dynamics in the bulk, we believe that very similar methods can be employed also to solve certain interesting stochastic deformations of the model. For example, in Ref. [19], a stochastic variant of our lattice dynamics has been proposed where the local propagator (6) is replaced by a Markov matrix

$$
U_\Gamma = \begin{pmatrix}
1 & 0 & 0 & 0 & 0 & 0 & 0 & 0 & 0 \\
0 & 0 & 0 & 1 & 0 & 0 & 0 & 0 & 0 \\
0 & 0 & 0 & 0 & 0 & 0 & 1 & 0 & 0 \\
0 & 1 & 0 & 0 & 0 & 0 & 0 & 0 & 0 \\
0 & 0 & 0 & 0 & 1 & 0 & 0 & 0 & 0 \\
0 & 0 & 0 & 0 & 0 & 1-\Gamma & 0 & \Gamma & 0 \\
0 & 0 & 1 & 0 & 0 & 0 & 0 & 0 & 0 \\
0 & 0 & 0 & 0 & 0 & \Gamma & 0 & 1-\Gamma & 0 \\
0 & 0 & 0 & 0 & 0 & 0 & 0 & 0 & 1
\end{pmatrix}, \tag{125}
$$

where $\Gamma \in [0, 1]$ is a probability of exchange of $+$ and $-$ particles upon their collision (scattering). Remarkably, even though $U_\Gamma$ in general no longer satisfies the braid relation (7), we find that certain nontrivial dynamical [19] and non-equilibrium steady state problems can still be exactly solved. In particular, we have empirical evidence that the stochastic deformation (125) of the steady state driven by stochastic boundaries can be written by a matrix product ansatz of exactly the same complexity (characterised by bond dimensions, or Schmidt ranks of all bi-partitions) as in the deterministic case ($\Gamma = 0$) studied in sect. 5.1. However, its precise analytic structure shall be left for future work.

## Acknowledgements

We warmly thank Katja Klobas for useful discussions.

**Funding information** Support from the Advanced grant 694544-OMNES of the European Research Council (ERC), and the Program P1-0402 of Slovenian Research Agency (ARRS) are gratefully acknowledged. VP acknowledges support from Deutsche Forschungsgemeinschaft, and from interdisciplinary UoC Forum "Classical and quantum dynamics of interacting particle systems".

# A   Details of the computations for the inhomogeneous quench

## A.1   Computation of the normalisation

The normalisation of the time-dependent matrix product state is defined as

$$Z_t = \langle l|\big((V_0 + V_+ + V_-)(W_0 + W_+ + W_-)\big)^t|r\rangle. \tag{126}$$

To evaluate explicitly this quantity, we first use the fact that

$$[V_0 + V_+ + V_-, W_0 + W_+ + W_-] = 0, \tag{127}$$

which can be readily checked using the explicit expression of the matrices (42). The normalisation can then be rewritten as

$$Z_t = \langle l|(W_0 + W_+ + W_-)^t(V_0 + V_+ + V_-)^t|r\rangle. \tag{128}$$

The boundary vectors $\langle l|$ and $|r\rangle$ (see explicit expressions in (43)) fulfill the following eigenvalue relations

$$\langle l|(W_0 + W_+ + W_-) = (\alpha_0 + \alpha)\langle l|, \qquad (V_0 + V_+ + V_-)|r\rangle = (\beta_0 + \beta)|r\rangle. \tag{129}$$

Those relations immediately yield the expression

$$Z_t = (\alpha_0 + \alpha)^t(\beta_0 + \beta)^t. \tag{130}$$

## A.2   Computation of the densities

The probability to observe a particle of species $\epsilon$ at site $j = 2k - t$, with $1 \le k \le t$, is given by the matrix product expression

$$n_\epsilon^e(j,t) = \frac{1}{Z_t}\langle l|T^{k-1}(V_0 + V_+ + V_-)W_\epsilon T^{t-k}|r\rangle, \tag{131}$$

where $T = (W_0 + W_+ + W_-)(V_0 + V_+ + V_-)$. Using the commutation property (127) and the eigenvectors relations (129) we can simplify the expression

$$n_\epsilon^e(j,t) = \frac{(\alpha_0 + \alpha)^{k-1}(\beta_0 + \beta)^{t-k}}{Z_t}\langle l|(V_0 + V_+ + V_-)^k W_\epsilon (W_0 + W_+ + W_-)^{t-k}|r\rangle. \tag{132}$$

The next step is to use the closure relation in the Fock space

$$\oint \frac{dz}{2i\pi z}|\underline{z}\rangle\langle\underline{1/z}| = 1, \tag{133}$$

and insert it to the left of $W_\epsilon$. Note that the integration contour includes the pole at $z = 0$ and excludes all others. Then using the fact that the coherent states $|\underline{z}\rangle$ and $\langle\underline{1/z}|$ are respectively eigenstates of the annihilation and creation operators $\mathfrak{a}$ and $\mathfrak{a}^\dagger$, we obtain

$$
\begin{aligned}
n_\epsilon^e(j,t) = {}& \frac{(\alpha_0 + \alpha)^{k-1}(\beta_0 + \beta)^{t-k}}{Z_t} \\
&\times \oint \frac{dz}{2i\pi z}\mathbf{e}_1\,(\beta_0\mathbb{I}_2 + z\alpha A + \beta B)^k\left(\alpha_\epsilon A + \frac{\beta_\epsilon}{z}B\right)\left(\alpha_0\mathbb{I}_2 + \alpha A + \frac{\beta}{z}B\right)^{t-k}\mathbf{v}.
\end{aligned}
\tag{134}
$$

The integrand can be evaluated using the triangular structure of matrices $A$ and $B$

$$\mathsf{e}_1 (\beta_0 \mathbb{I}_2 + z\alpha A + \beta B)^k \left( \alpha_\epsilon A + \frac{\beta_\epsilon}{z} B \right) \left( \alpha_0 \mathbb{I}_2 + \alpha A + \frac{\beta}{z} B \right)^{t-k} \mathsf{v} =$$
$$\frac{\beta\alpha_\epsilon - \alpha\beta_\epsilon}{\beta - z\alpha} (\beta_0 + z\alpha)^k \left( \frac{\beta}{z} + \alpha_0 \right)^{t-k}$$
$$+ \frac{\frac{\beta\beta_\epsilon}{z}(\beta_0 + \beta)^k \left( \frac{\beta}{z} + \alpha_0 \right)^{t-k} - z\alpha\alpha_\epsilon (\alpha_0 + \alpha)^{t-k} (\beta_0 + z\alpha)^k}{\beta - z\alpha}. \tag{135}$$

The residue at $z = 0$ corresponding to the first of the two previous terms is equal to

$$\frac{\beta\alpha_\epsilon - \alpha\beta_\epsilon}{(t-k)!} \frac{d^{t-k}}{dz^{t-k}} \frac{(\beta_0 + z\alpha)^k(\beta + z\alpha_0)^{t-k}}{\beta - z\alpha} \bigg|_{z=0} =$$
$$= \frac{(\beta\alpha_\epsilon - \alpha\beta_\epsilon)\beta_0^k \alpha^{t-k}}{\beta} \sum_{n=0}^{t-k} \sum_{i=0}^{\min(n,k)} \binom{k}{i}\binom{t-k}{n-i}\left(\frac{\alpha_0}{\alpha}\right)^{n-i}\left(\frac{\beta}{\beta_0}\right)^i, \tag{136}$$

whereas the residue at $z = 0$ corresponding to the second term is equal to

$$\frac{\beta\beta_\epsilon(\beta_0 + \beta)^k}{(t-k+1)!} \frac{d^{t-k+1}}{dz^{t-k+1}} \frac{(\beta + z\alpha_0)^{t-k}}{\beta - z\alpha} \bigg|_{z=0} = \frac{\alpha\beta_\epsilon}{\beta}(\beta_0 + \beta)^k(\alpha_0 + \alpha)^{t-k}. \tag{137}$$

Gathering these results together we end up with the following expression

$$n_\epsilon^{\mathrm{e}}(j,t) = \frac{\beta_\epsilon}{\beta} \frac{\alpha}{\alpha_0 + \alpha} + \frac{(\beta\alpha_\epsilon - \alpha\beta_\epsilon)\beta_0^k \alpha^{t-k}}{\beta(\beta_0 + \beta)^k(\alpha_0 + \alpha)^{t-k+1}} \sum_{n=0}^{t-k} \sum_{i=0}^{\min(n,k)} \binom{k}{i}\binom{t-k}{n-i}\left(\frac{\alpha_0}{\alpha}\right)^{n-i}\left(\frac{\beta}{\beta_0}\right)^i. \tag{138}$$

Finally, using the fact that $\alpha_0 + \alpha = 1$ and $\beta_0 + \beta = 1$, the expression reduces to

$$n_\epsilon^{\mathrm{e}}(j,t) = \beta_\epsilon\frac{\alpha}{\beta} + \left(\alpha_\epsilon - \beta_\epsilon\frac{\alpha}{\beta}\right)(1-\beta)^k\alpha^{t-k} \sum_{n=0}^{t-k} \sum_{i=0}^{\min(n,k)} \binom{k}{i}\binom{t-k}{n-i}\left(\frac{1-\alpha}{\alpha}\right)^{n-i}\left(\frac{\beta}{1-\beta}\right)^i. \tag{139}$$

Note that we also have an alternative expression for the density, if we do not evaluate explicitly the residue (136)

$$n_\epsilon^{\mathrm{e}}(j,t) = \beta_\epsilon\frac{\alpha}{\beta} + \oint \frac{dz}{2i\pi z} \frac{\beta\alpha_\epsilon - \alpha\beta_\epsilon}{\beta - z\alpha}(1 - \beta + z\alpha)^k \left(\frac{\beta}{z} + 1 - \alpha\right)^{t-k}, \tag{140}$$

where the contour of the integral encircles the pole at $z = 0$ and excludes the pole at $z = \beta/\alpha$. The probability to observe a particle of species $\epsilon$ at site $j = 2k + 1 - t$, with $0 \le k \le t - 1$, is given by the matrix product expression

$$n_\epsilon^{\mathrm{o}}(j,t) = \frac{1}{Z_t}\langle l|T^k V_\epsilon(W_0 + W_+ + W_-)T^{t-k-1}|r\rangle. \tag{141}$$

A very similar computation yields

$$n_\epsilon^{\mathrm{o}}(j,t) = \frac{\beta_\epsilon}{\beta_0 + \beta} + \frac{(\beta\alpha_\epsilon - \alpha\beta_\epsilon)\beta_0^k \alpha^{t-k-1}}{(\beta_0 + \beta)^{k+1}(\alpha_0 + \alpha)^{t-k}} \sum_{n=0}^{t-k-1} \sum_{i=0}^{\min(n,k)} \binom{k}{i}\binom{t-k}{n-i}\left(\frac{\alpha_0}{\alpha}\right)^{n-i}\left(\frac{\beta}{\beta_0}\right)^i. \tag{142}$$

Using the fact that $\alpha_0 + \alpha = 1$ and $\beta_0 + \beta = 1$, the expression reduces to

$$n_\epsilon^{\mathrm{o}}(j,t) = \beta_\epsilon + \left(\frac{\beta}{\alpha}\alpha_\epsilon - \beta_\epsilon\right)(1-\beta)^k\alpha^{t-k} \sum_{n=0}^{t-k-1} \sum_{i=0}^{\min(n,k)} \binom{k}{i}\binom{t-k}{n-i}\left(\frac{1-\alpha}{\alpha}\right)^{n-i}\left(\frac{\beta}{1-\beta}\right)^i. \tag{143}$$

As before, we also have a contour integral expression

$$n_\epsilon^\circ(j,t) = \beta_\epsilon + \oint \frac{dz}{2i\pi} \frac{\beta\alpha_\epsilon - \alpha\beta_\epsilon}{\beta - z\alpha} (1 - \beta + z\alpha)^k \left(\frac{\beta}{z} + 1 - \alpha\right)^{t-k}. \tag{144}$$

# B  Details of the computations for the local quench

## B.1  Computation of the normalisation

The normalisation of the time-dependent matrix product state is defined as

$$Z_t = (\langle l|_0 + \langle l|_+ + \langle l|_-) \, T^{t-1} (W_0 + W_+ + W_-)|r\rangle, \tag{145}$$

where $T = (W_0 + W_+ + W_-)(V_0 + V_+ + V_-)$. To evaluate this quantity explicitly we first use the fact that

$$[V_0 + V_+ + V_-, W_0 + W_+ + W_-] = 0, \tag{146}$$

which can be readily checked using the explicit expression of the matrices (67). The normalization can then be rewritten as

$$Z_t = (\langle l|_0 + \langle l|_+ + \langle l|_-)(W_0 + W_+ + W_-)^t (V_0 + V_+ + V_-)^{t-1}|r\rangle. \tag{147}$$

The boundary vectors $\langle l|_\tau$ and $|r\rangle$ (see explicit expressions in (68)) fulfill the following eigenvalue relations

$$(\langle l|_0 + \langle l|_+ + \langle l|_-)(W_0 + W_+ + W_-) = (\rho_0 + \rho)(\langle l|_0 + \langle l|_+ + \langle l|_-),$$
$$(V_0 + V_+ + V_-)|r\rangle = (\rho_0 + \rho)|r\rangle. \tag{148}$$

We have also the scalar product expression

$$(\langle l|_0 + \langle l|_+ + \langle l|_-) \cdot |r\rangle = \lambda_0 + \lambda. \tag{149}$$

Those relations lead immediately to the expression

$$Z_t = (\rho_0 + \rho)^{2t-1}(\lambda_0 + \lambda). \tag{150}$$

## B.2  Computation of the densities

The probability to observe a particle of species $\epsilon$ at site $j = 2k - t$, with $1 \le k \le t$, is expressed within the matrix product framework as

$$n_\epsilon^e(j,t) = \frac{1}{Z_t} (\langle l|_0 + \langle l|_+ + \langle l|_-) \, T^{k-1} W_\epsilon T^{t-k}|r\rangle. \tag{151}$$

Using the commutation property (146) and the eigenvectors properties (148) the expression is simplified into

$$n_\epsilon^e(j,t) = \frac{(\rho_0 + \rho)^{t-1}}{Z_t} (\langle l|_0 + \langle l|_+ + \langle l|_-)(V_0 + V_+ + V_-)^{k-1} W_\epsilon (W_0 + W_+ + W_-)^{t-k}|r\rangle. \tag{152}$$

Then using the explicit expression of the matrices (67) and of the boundary vectors (68) and the properties $V_\tau \mathsf{P} = V_\tau$, $W_\tau \mathsf{P} = W_\tau$ and $\langle l|_\tau \mathsf{P} = \langle l|_\tau$, $\tau = \pm, 0$, we can get rid of the projector $\mathsf{P}$ in the matrix product and we obtain

$$n_\epsilon^e(j,t) = \left(\rho\langle 1|, \ \lambda\langle 0| + \lambda_0\langle\underline{1}|\right) \begin{pmatrix} \rho_0 + \rho\mathfrak{a} & \lambda \\ 0 & \rho_0 + \rho \end{pmatrix}^{k-1} \begin{pmatrix} \rho_\epsilon & \lambda_\epsilon\mathfrak{a}^\dagger \\ 0 & \rho_\epsilon\mathfrak{a}^\dagger \end{pmatrix} \begin{pmatrix} \rho_0 + \rho & \lambda\mathfrak{a}^\dagger \\ 0 & \rho_0 + \rho\mathfrak{a}^\dagger \end{pmatrix}^{t-k} \begin{pmatrix} 0 \\ |0\rangle \end{pmatrix}. \tag{153}$$

The next step is to use the closure relation in the Fock space

$$\oint \frac{dz}{2i\pi z} |\underline{z}\rangle \langle \underline{1/z}| = 1,$$ (154)

and insert it once between[2] the operators $\mathfrak{a}$ and $\mathfrak{a}^\dagger$. Note that the integration contour includes the pole at $z = 0$ and excludes all others. Then using the fact that the coherent states $|\underline{z}\rangle$ and $\langle \underline{1/z}|$ are respectively eigenstates of the annihilation and creation operators $\mathfrak{a}$ and $\mathfrak{a}^\dagger$, we obtain

$$n_\epsilon^e(j,t) = \oint \frac{dz}{2i\pi z} \big(\rho z, \ \lambda + \frac{\lambda_0}{1-z}\big) \begin{pmatrix} \rho_0 + \rho z & \lambda \\ 0 & \rho_0 + \rho \end{pmatrix}^{k-1} \begin{pmatrix} \rho_\epsilon & \frac{\lambda_\epsilon}{z} \\ 0 & \frac{\rho_\epsilon}{z} \end{pmatrix} \begin{pmatrix} \rho_0 + \rho & \frac{\lambda}{z} \\ 0 & \rho_0 + \frac{\rho}{z} \end{pmatrix}^{t-k} \begin{pmatrix} 0 \\ 1 \end{pmatrix}.$$ (155)

The integrand can be further simplified using the triangular structure of the matrices. It reads

$$(\rho \lambda_\epsilon - \lambda \rho_\epsilon)(\rho_0 + \rho z)^{k-1} \big(\rho_0 + \frac{\rho}{z}\big)^{t-k} + \frac{(\lambda_0 + \lambda)\rho_\epsilon (\rho_0 + \rho)^{k-1} \big(\rho_0 + \frac{\rho}{z}\big)^{t-k}}{z(1-z)}$$
$$- \frac{z \lambda \rho_\epsilon (\rho_0 + \rho z)^{k-1}(\rho_0 + \rho)^{t-k}}{1-z}.$$ (156)

A straightforward computation yields the following results. The residue at $z = 0$ corresponding to the third term is equal to 0. The residue corresponding to the second term at $z = 0$ is equal to $\rho_\epsilon(\lambda_0 + \lambda)(\rho_0 + \rho)^{t-1}$, whereas the one corresponding to the first term is given by

$$(\rho \lambda_\epsilon - \lambda \rho_\epsilon) \sum_{l=0}^{\min(k-1, t-k)} \binom{k-1}{l}\binom{t-k}{l} \rho^{2l} \rho_0^{t-1-2l}.$$ (157)

Gathering the results together we obtain an expression of the density

$$n_\epsilon^e(j,t) = \frac{\rho_\epsilon}{\rho_0 + \rho} + \frac{\rho \lambda_\epsilon - \lambda \rho_\epsilon}{(\lambda_0 + \lambda)(\rho_0 + \rho)^t} \sum_{l=0}^{\min(k-1, t-k)} \binom{k-1}{l}\binom{t-k}{l} \rho^{2l} \rho_0^{t-1-2l}.$$ (158)

Using the fact that $\rho_0 + \rho = 1$ and $\lambda_0 + \lambda = 1$ the expression reduces to

$$n_\epsilon^e(j,t) = \rho_\epsilon + (\rho \lambda_\epsilon - \lambda \rho_\epsilon) \sum_{l=0}^{\min(k-1, t-k)} \binom{k-1}{l}\binom{t-k}{l} \rho^{2l}(1-\rho)^{t-1-2l}.$$ (159)

Note that we have also an alternative expression of the density if we do not evaluate explicitly the residue associated to the first term in (156)

$$n_\epsilon^e(j,t) = \rho_\epsilon + (\rho \lambda_\epsilon - \lambda \rho_\epsilon) \oint \frac{dz}{2i\pi z} (1-\rho+\rho z)^{k-1} \big(1 - \rho + \frac{\rho}{z}\big)^{t-k},$$ (160)

where the contour of the integral encircles the pole at $z = 0$. The probability to observe a particle of species $\epsilon$ at site $j = 2k + 1 - t$, with $1 \le k \le t - 1$, is given by the matrix product expression

$$n_\epsilon^o(j,t) = \frac{1}{Z_t} (\langle l|_0 + \langle l|_+ + \langle l|_-) T^{k-1}(W_0 + W_+ + W_-)V_\epsilon(W_0 + W_+ + W_-)T^{t-k-1}|r\rangle.$$ (161)

---

[2]One can indeed observe in equation (153) that the operators $\mathfrak{a}$ are all located on the left side and the operators $\mathfrak{a}^\dagger$ are all located on the right side of the expression.

A very similar computation yields to

$$n_\epsilon^o(j,t) = \frac{\rho_\epsilon}{\rho_0 + \rho} + \frac{\rho\lambda_\epsilon - \lambda\rho_\epsilon}{(\lambda_0 + \lambda)(\rho_0 + \rho)^t} \sum_{l=0}^{\min(k-1,t-k-1)} \binom{k-1}{l}\binom{t-k}{l+1}\rho^{2l+1}\rho_0^{t-2-2l}, \quad (162)$$

which reduces to

$$n_\epsilon^o(j,t) = \rho_\epsilon + (\rho\lambda_\epsilon - \lambda\rho_\epsilon) \sum_{l=0}^{\min(k-1,t-k-1)} \binom{k-1}{l}\binom{t-k}{l+1}\rho^{2l+1}(1-\rho)^{t-2-2l}. \quad (163)$$

If we do not evaluate explicitly the residue, we obtain an alternative expression of the densities,

$$n_\epsilon^o(j,t) = \rho_\epsilon + (\rho\lambda_\epsilon - \lambda\rho_\epsilon) \oint \frac{dz}{2i\pi}(1-\rho+\rho z)^{k-1}\left(1-\rho+\frac{\rho}{z}\right)^{t-k}. \quad (164)$$

Finally we have the particular case of the density of species $\epsilon$ at site $j = -t + 1$. The corresponding matrix product expression is

$$n_\epsilon^o(-t+1,t) = \frac{1}{Z_t}\langle l|_\epsilon T^{t-1}(W_0 + W_+ + W_-)|r\rangle. \quad (165)$$

Using the commutation property (146), the right eigenvector property in (148) we obtain

$$n_\epsilon^o(-t+1,t) = \frac{(\rho_0+\rho)^{t-1}}{Z_t}\langle l|_\epsilon(W_0 + W_+ + W_-)^t|r\rangle. \quad (166)$$

Then using recursively the relations

$$\begin{aligned}
\langle l|_\epsilon(W_0 + W_+ + W_-) &= \rho_0\langle l|_\epsilon + \rho_\epsilon(\langle l|_+ + \langle l|_-), \\
(\langle l|_+ + \langle l|_-)(W_0 + W_+ + W_-) &= (\rho_0+\rho)(\langle l|_+ + \langle l|_-),
\end{aligned} \quad (167)$$

we get

$$\begin{aligned}
\langle l|_\epsilon(W_0 + W_+ + W_-)^t &= \rho_0^t\langle l|_\epsilon + \rho_\epsilon\left(\sum_{l=0}^{t-1}\rho_0^l(\rho_0+\rho)^{t-l-1}\right)(\langle l|_+ + \langle l|_-) \\
&= \rho_0^t\langle l|_\epsilon + \rho_\epsilon\frac{(\rho_0+\rho)^t - \rho_0^t}{\rho}(\langle l|_+ + \langle l|_-).
\end{aligned} \quad (168)$$

The scalar product relations

$$\langle l|_\epsilon \cdot |r\rangle = \lambda_\epsilon, \qquad (\langle l|_+ + \langle l|_-) \cdot |r\rangle = \lambda \quad (169)$$

yield the final expression

$$n_\epsilon^o(-t+1,t) = \frac{\rho_0^t(\rho\lambda_\epsilon - \lambda\rho_\epsilon)}{\rho(\lambda_0+\lambda)(\rho_0+\rho)^t} + \frac{\rho_\epsilon\lambda}{\rho(\lambda_0+\lambda)}. \quad (170)$$

Taking into account the fact that $\rho_0 + \rho = 1$ and $\lambda_0 + \lambda = 1$, the formula reduces to

$$n_\epsilon^o(-t+1,t) = \frac{(1-\rho)^t(\rho\lambda_\epsilon - \lambda\rho_\epsilon)}{\rho} + \frac{\rho_\epsilon\lambda}{\rho}. \quad (171)$$

## C Details of the computations for the boundary driven model

### C.1 Computation of the normalisation

The normalisation of the stationary distribution is defined as

$$Z_L = \left( \sum_{\tau \in \{0,+,-\}} \langle l|_\tau \right) \left( (V_0 + V_+ + V_-)(W_0 + W_+ + W_-) \right)^{\frac{L-3}{2}} \left( \sum_{\tau,\tau' \in \{0,+,-\}} |rr\rangle_{\tau\tau'} \right). \quad (172)$$

The first step to compute this quantity is to remark the commutation property

$$[V_0 + V_+ + V_-, W_0 + W_+ + W_-] = 0, \quad (173)$$

which can be directly checked using the explicit representation (93). We can thus rewrite

$$Z_L = \left( \sum_{\tau \in \{0,+,-\}} \langle l|_\tau \right) (W_0 + W_+ + W_-)^{\frac{L-3}{2}} (V_0 + V_+ + V_-)^{\frac{L-3}{2}} \cdot \left( \sum_{\tau,\tau' \in \{0,+,-\}} |rr\rangle_{\tau\tau'} \right). \quad (174)$$

We have the following expressions

$$V_0 + V_+ + V_- = \alpha\big(\beta_0 \mathbb{I}_2 + \beta B\big) \otimes 1 \otimes (1 + \mathfrak{a}^\dagger) + \alpha A \otimes \mathfrak{a} \otimes \big(\beta_0 \mathfrak{s} + \beta(1 + \mathfrak{a})\big),$$

$$W_0 + W_+ + W_- = \beta\big(\alpha_0 \mathbb{I}_2 + \alpha A\big) \otimes (1 + \mathfrak{a}) \otimes 1 + \beta B \otimes \big(\alpha_0 \mathfrak{s} + \alpha(1 + \mathfrak{a}^\dagger)\big) \otimes \mathfrak{a}^\dagger,$$

$$\sum_{\tau \in \{0,+,-\}} \langle l|_\tau = (\alpha + \alpha_0) e_1^t \otimes \langle 0| \otimes \langle \underline{\beta/\beta_0}|, \quad (175)$$

$$\sum_{\tau,\tau' \in \{0,+,-\}} |rr\rangle_{\tau\tau'} = (\alpha + \alpha_0)(\beta + \beta_0)\beta v \otimes |\underline{\alpha/\alpha_0}\rangle \otimes (1 + \mathfrak{a}^\dagger)|0\rangle$$

$$+ \alpha\beta(\beta + \beta_0)\left(1 + \frac{\alpha}{\alpha_0}\right) e_1 \otimes |\underline{\alpha/\alpha_0}\rangle \otimes |0\rangle.$$

Using these expressions it is straightforward to see that the actions of the coherent states $\langle \underline{\beta/\beta_0}|$ (in the third tensor component) and $|\underline{\alpha/\alpha_0}\rangle$ (in the second tensor component) can be evaluated because of the relations (38). We thus have

$$e_1^t \otimes \langle 0| \otimes \langle \underline{\beta/\beta_0}|(W_0 + W_+ + W_-)^n (V_0 + V_+ + V_-)^n v \otimes |\underline{\alpha/\alpha_0}\rangle \otimes (1 + \mathfrak{a}^\dagger)|0\rangle$$

$$= \alpha^n \beta^n (\alpha + \alpha_0)^n (\beta + \beta_0)^n \left(1 + \frac{\beta}{\beta_0}\right) e_1^t \left(\mathbb{I}_2 + \frac{\alpha}{\alpha_0}A + \frac{\beta}{\beta_0}B\right)^{2n} v \quad (176)$$

and

$$e_1^t \otimes \langle 0| \otimes \langle \underline{\beta/\beta_0}|(W_0 + W_+ + W_-)^n (V_0 + V_+ + V_-)^n e_1 \otimes |\underline{\alpha/\alpha_0}\rangle \otimes |0\rangle$$

$$= \alpha^n \beta^n (\alpha + \alpha_0)^n (\beta + \beta_0)^n e_1^t \left(\mathbb{I}_2 + \frac{\alpha}{\alpha_0}A + \frac{\beta}{\beta_0}B\right)^{2n} e_1. \quad (177)$$

Using the explicit expression

$$\mathbb{I}_2 + \frac{\alpha}{\alpha_0}A + \frac{\beta}{\beta_0}B = \begin{pmatrix} 1 + \frac{\alpha}{\alpha_0} & \frac{\beta}{\beta_0} \\ 0 & 1 + \frac{\beta}{\beta_0} \end{pmatrix}, \quad (178)$$

we can compute

$$e_1^t \left(\mathbb{I}_2 + \frac{\alpha}{\alpha_0}A + \frac{\beta}{\beta_0}B\right)^{2n} e_1 = \left(1 + \frac{\alpha}{\alpha_0}\right)^{2n} \quad (179)$$

and

$$\mathrm{e}_1^t \left( \mathbb{I}_2 + \frac{\alpha}{\alpha_0} A + \frac{\beta}{\beta_0} B \right)^{2n} \mathsf{v} = \frac{\frac{\alpha}{\alpha_0} \left( 1 + \frac{\alpha}{\alpha_0} \right)^{2n} - \frac{\beta}{\beta_0} \left( 1 + \frac{\beta}{\beta_0} \right)^{2n}}{\frac{\alpha}{\alpha_0} - \frac{\beta}{\beta_0}}. \tag{180}$$

Gathering all these results together, the normalisation can be explicitly evaluated

$$Z_L = \frac{\frac{\alpha}{\alpha_0} \left( 1 + \frac{\alpha}{\alpha_0} \right)^{L-2} - \frac{\beta}{\beta_0} \left( 1 + \frac{\beta}{\beta_0} \right)^{L-2}}{\frac{\alpha}{\alpha_0} - \frac{\beta}{\beta_0}} \alpha^{\frac{L-3}{2}} \beta^{\frac{L-1}{2}} (\alpha + \alpha_0)^{\frac{L+1}{2}} (\beta + \beta_0)^{\frac{L-1}{2}}. \tag{181}$$

Taking into account that $\alpha + \alpha_0 = 1$ and $\beta + \beta_0 = 1$, the expression reduces to

$$Z_L = \frac{\alpha(1-\beta)^{L-1} - \beta(1-\alpha)^{L-1}}{\alpha - \beta} \frac{\alpha^{\frac{L-3}{2}} \beta^{\frac{L-1}{2}}}{(1-\alpha)^{L-2}(1-\beta)^{L-2}}. \tag{182}$$

## C.2  Computation of the particle currents

The particle current associated to the species $\epsilon = \pm$ is expressed as (using again the commutation property (173))

$$J_\epsilon = \frac{1}{Z_L} (\langle l|_\epsilon V_0 - \langle l|_0 V_\epsilon)(W_0 + W_+ + W_-)^{\frac{L-3}{2}} (V_0 + V_+ + V_-)^{\frac{L-5}{2}} \left( \sum_{\tau,\tau' \in \{0,+,-\}} |rr\rangle_{\tau\tau'} \right). \tag{183}$$

We have the relation

$$\langle l|_\epsilon V_0 - \langle l|_0 V_\epsilon = \alpha_\epsilon \alpha(\beta + \beta_0)\mathrm{e}_1^t \otimes \langle 0| \otimes \langle \underline{\beta/\beta_0}| + \alpha_\epsilon \beta_0(\alpha + \alpha_0)\mathrm{e}_1^t \otimes \langle 1| \otimes \langle 0|$$
$$- \alpha_0 \alpha_\epsilon(\beta + \beta_0)\mathrm{e}_1^t \otimes \langle 1| \otimes \langle \underline{\beta/\beta_0}| - \alpha\alpha_0\beta_\epsilon \left( 1 + \frac{\beta}{\beta_0} \right) \mathrm{e}_2^t \otimes \langle 0| \otimes \langle \underline{\beta/\beta_0}|. \tag{184}$$

Similarly to what has been done for the computation of the normalisation, we can evaluate the action of the coherent states $|\alpha/\alpha_0\rangle$ and $\langle \beta/\beta_0|$. It allows one to realize that the contributions of $\alpha_\epsilon \alpha(\beta + \beta_0)\mathrm{e}_1^t \otimes \langle 0| \otimes \langle \underline{\beta/\beta_0}|$ and of $-\alpha_0\alpha_\epsilon(\beta + \beta_0)\mathrm{e}_1^t \otimes \langle 1| \otimes \langle \underline{\beta/\beta_0}|$ (appearing in (184)) in the computation of the current cancel with each other because $\overline{\langle 0| \cdot |\alpha/\alpha_0\rangle} = 1$ and $\langle 1| \cdot |\underline{\alpha/\alpha_0}\rangle = \alpha/\alpha_0$. Concerning the remaining terms, we have

$$\mathrm{e}_2^t \otimes \langle 0| \otimes \langle \underline{\beta/\beta_0}|(W_0 + W_+ + W_-)^n (V_0 + V_+ + V_-)^{n-1} \mathrm{e}_1 \otimes |\underline{\alpha/\alpha_0}\rangle \otimes |0\rangle$$
$$= \alpha^{n-1} \beta^n (\alpha + \alpha_0)^n (\beta + \beta_0)^{n-1} \mathrm{e}_2^t \left( \mathbb{I}_2 + \frac{\alpha}{\alpha_0} A + \frac{\beta}{\beta_0} B \right)^{2n-1} \mathrm{e}_1 = 0, \tag{185}$$

and

$$\mathrm{e}_2^t \otimes \langle 0| \otimes \langle \underline{\beta/\beta_0}|(W_0 + W_+ + W_-)^n (V_0 + V_+ + V_-)^{n-1} \mathsf{v} \otimes |\underline{\alpha/\alpha_0}\rangle \otimes (1 + \mathfrak{a}^\dagger)|0\rangle$$
$$= \alpha^{n-1} \beta^n (\alpha + \alpha_0)^n (\beta + \beta_0)^{n-1} \left( 1 + \frac{\beta}{\beta_0} \right) \mathrm{e}_2^t \left( \mathbb{I}_2 + \frac{\alpha}{\alpha_0} A + \frac{\beta}{\beta_0} B \right)^{2n-1} \mathsf{v}$$
$$= \alpha^{n-1} \beta^n (\alpha + \alpha_0)^n (\beta + \beta_0)^{n-1} \left( 1 + \frac{\beta}{\beta_0} \right)^{2n}. \tag{186}$$

The contribution of the term $\alpha_\epsilon \beta_0(\alpha + \alpha_0)\mathrm{e}_1^t \otimes \langle 1| \otimes \langle 0|$ of (184) is a bit more involved to compute because we do not have the coherent state $\langle \underline{\beta/\beta_0}|$ anymore. We are going to show that

$$\mathrm{e}_1^t \otimes \langle 1| \otimes \langle 0|(W_0 + W_+ + W_-)^n (V_0 + V_+ + V_-)^{n-1} \left( \sum_{\tau,\tau' \in \{0,+,-\}} |rr\rangle_{\tau\tau'} \right)$$
$$= \alpha^n \beta^{n+1} (\alpha + \alpha_0)^n (\beta + \beta_0)^n \left( 1 + \frac{\alpha}{\alpha_0} \right)^{2n+1}. \tag{187}$$

Using (184), (185), (186) and (187) it is then straightforward to show that

$$J_\epsilon = \frac{\beta_0 \alpha_\epsilon \left(1 + \frac{\alpha}{\alpha_0}\right)^{L-2} - \alpha_0 \beta_\epsilon \left(1 + \frac{\beta}{\beta_0}\right)^{L-2}}{\beta_0 \alpha \left(1 + \frac{\alpha}{\alpha_0}\right)^{L-2} - \alpha_0 \beta \left(1 + \frac{\beta}{\beta_0}\right)^{L-2}} \frac{\beta_0 \alpha - \alpha_0 \beta}{(\alpha + \alpha_0)(\beta + \beta_0)}. \tag{188}$$

Taking into account that $\alpha + \alpha_0 = 1$ and $\beta + \beta_0 = 1$, the expression reduces to

$$J_\epsilon = \frac{\alpha_\epsilon (1-\beta)^{L-1} - \beta_\epsilon (1-\alpha)^{L-1}}{\alpha (1-\beta)^{L-1} - \beta (1-\alpha)^{L-1}} (\alpha - \beta). \tag{189}$$

We now come back to the proof of equation (187). The first step is to observe that the action of the coherent state $|\overline{\alpha/\alpha_0}\rangle$ in the second tensor component of the right boundary vectors can be evaluated, similarly as in the previous computations. The action of the vector $\langle 0|$ in the third tensor component of the left vector can be directly evaluated on $(W_0 + W_+ + W_-)^n$. This last step kills the term involving the matrix $B$ in $W_0 + W_+ + W_-$ (see equation (175)), so that $\mathsf{e}_1^t$ is a left eigenvector of the remaining terms ($\mathsf{e}_1^t$ is a left eigenvector of the matrix $A$). Applying all these manipulations we end up with

$$\mathsf{e}_1^t \otimes \langle 1| \otimes \langle 0|(W_0 + W_+ + W_-)^n (V_0 + V_+ + V_-)^{n-1} \left( \sum_{\tau, \tau' \in \{0,+,-\}} |rr\rangle_{\tau\tau'} \right)$$

$$= \alpha^n \beta^{n+1} (\beta + \beta_0)(\alpha + \alpha_0)^n \left(1 + \frac{\alpha}{\alpha_0}\right)^{n+1}$$

$$\times \mathsf{e}_1^t \otimes \langle 0| \left[ (\beta_0 \mathbb{I}_2 + \beta B) \otimes (1 + \mathfrak{a}^\dagger) + \frac{\alpha}{\alpha_0} A \otimes (\beta_0 \mathfrak{s} + \beta(1+\mathfrak{a})) \right]^{n-1} \left( \mathsf{v} \otimes (1 + \mathfrak{a}^\dagger)|0\rangle + \frac{\alpha}{\alpha_0} \mathsf{e}_1 \otimes |0\rangle \right). \tag{190}$$

The last point left to complete the proof is to show the following relation

$$\mathsf{e}_1^t \otimes \langle 0| \left[ (\beta_0 \mathbb{I}_2 + \beta B) \otimes (1 + \mathfrak{a}^\dagger) + \frac{\alpha}{\alpha_0} A \otimes (\beta_0 \mathfrak{s} + \beta(1+\mathfrak{a})) \right]^{n-1} \left( \mathsf{v} \otimes (1 + \mathfrak{a}^\dagger)|0\rangle + \frac{\alpha}{\alpha_0} \mathsf{e}_1 \otimes |0\rangle \right)$$

$$= (\beta + \beta_0)^{n-1} \left(1 + \frac{\alpha}{\alpha_0}\right)^n. \tag{191}$$

Using the triangular structure of the matrices $A$ and $B$ we can evaluate the matrix product in the finite dimensional auxiliary space as follows

$$\mathsf{e}_1^t \otimes \langle 0| \left[ (\beta_0 \mathbb{I}_2 + \beta B) \otimes (1 + \mathfrak{a}^\dagger) + \frac{\alpha}{\alpha_0} A \otimes (\beta_0 \mathfrak{s} + \beta(1+\mathfrak{a})) \right]^{n-1} \left( \mathsf{v} \otimes (1 + \mathfrak{a}^\dagger)|0\rangle + \frac{\alpha}{\alpha_0} \mathsf{e}_1 \otimes |0\rangle \right)$$

$$= \sum_{k=0}^{n-2} \beta(\beta + \beta_0)^{n-k-2} \langle 0| \left[ \frac{\alpha}{\alpha_0} (\beta_0 \mathfrak{s} + \beta(1+\mathfrak{a})) + \beta_0 (1 + \mathfrak{a}^\dagger) \right]^k (1 + \mathfrak{a}^\dagger)^{n-k}|0\rangle$$

$$+ \langle 0| \left[ \frac{\alpha}{\alpha_0} (\beta_0 \mathfrak{s} + \beta(1+\mathfrak{a})) + \beta_0 (1 + \mathfrak{a}^\dagger) \right]^{n-1} \left[ (1 + \mathfrak{a}^\dagger)|0\rangle + \frac{\alpha}{\alpha_0} |0\rangle \right]. \tag{192}$$

To compute the expression above, we introduce the ket and bra vectors

$$|v(z)\rangle = \sum_{l=0}^{+\infty} \left[ (z - \delta)(\gamma z)^l - \left(\frac{1}{z} - \delta\right)\left(\frac{\gamma}{z}\right)^l \right] |l\rangle,$$

$$\langle w(z)| = \sum_{l=0}^{+\infty} \left[ \left(\frac{1}{z} - \delta\right)\left(\frac{1}{\gamma z}\right)^l - (z - \delta)\left(\frac{z}{\gamma}\right)^l \right] \langle l|, \tag{193}$$

where $\gamma = \sqrt{\frac{\alpha_0 \beta_0}{\alpha \beta}}$ and $\delta = \sqrt{\frac{\alpha \beta_0}{\alpha_0 \beta}}$. Note that they can be conveniently expressed using coherent states

$$|v(z)\rangle = (z - \delta)\left|\underline{\gamma z}\right\rangle - \left(\frac{1}{z} - \delta\right)\left|\underline{\gamma/z}\right\rangle \tag{194}$$

and similarly for $\langle w(z)|$. They are the right and left eigenvectors of the following operator

$$\left[\frac{\alpha}{\alpha_0}\left(\beta_0 \mathfrak{s} + \beta(1 + \mathfrak{a})\right) + \beta_0(1 + \mathfrak{a}^\dagger)\right]|v(z)\rangle = \left[\beta_0 + \frac{\alpha}{\alpha_0}\beta + \sqrt{\frac{\alpha}{\alpha_0}\beta\beta_0}\left(z + \frac{1}{z}\right)\right]|v(z)\rangle,$$

$$\langle w(z)|\left[\frac{\alpha}{\alpha_0}\left(\beta_0 \mathfrak{s} + \beta(1 + \mathfrak{a})\right) + \beta_0(1 + \mathfrak{a}^\dagger)\right] = \left[\beta_0 + \frac{\alpha}{\alpha_0}\beta + \sqrt{\frac{\alpha}{\alpha_0}\beta\beta_0}\left(z + \frac{1}{z}\right)\right]\langle w(z)|. \tag{195}$$

They satisfy also the following closure relation

$$\oint \frac{dz}{2\pi i z}\frac{1}{2}\frac{1}{(z - \delta)\left(\frac{1}{z} - \delta\right)}|v(z)\rangle\langle w(z)| = 1, \tag{196}$$

where the integration contour includes poles at $z = 0$ and at $z = \delta$ and excludes any other. We insert the closure relation in (192) and then use the eigenvalue relations (195) and the fact that

$$\langle w(z)|(1 + \mathfrak{a}^\dagger)^j = \left(\frac{1}{z} - \delta\right)\left(1 + \frac{1}{\gamma z}\right)^j\left\langle\underline{1/(\gamma z)}\right| - (z - \delta)\left(1 + \frac{z}{\gamma}\right)^j\left\langle\underline{z/\gamma}\right|. \tag{197}$$

The summation in (192) is then simply a geometric summation in the integrand. After simplifications we are left with

$$\mathrm{e}_1^t \otimes \langle 0|\left[(\beta_0 \mathbb{I}_2 + \beta B) \otimes (1 + \mathfrak{a}^\dagger) + \frac{\alpha}{\alpha_0} A \otimes (\beta_0 \mathfrak{s} + \beta(1 + \mathfrak{a}))\right]^{n-1}\left(v \otimes (1 + \mathfrak{a}^\dagger)|0\rangle + \frac{\alpha}{\alpha_0}\mathrm{e}_1 \otimes |0\rangle\right)$$

$$= (\beta + \beta_0)^{n-1}\oint \frac{dz}{2\pi i}\frac{1}{2}\frac{z^2 - 1}{z(z - \delta)(1 - z\delta)}\left[\frac{1}{z}\left(1 + \frac{1}{\gamma z}\right)^n - z\left(1 + \frac{z}{\gamma}\right)^n\right]. \tag{198}$$

The residue at $z = \delta$ of the integrand can be easily computed (it is a simple pole) and is equal to

$$\frac{1}{2}\left[\left(1 + \frac{\alpha}{\alpha_0}\right)^n - \frac{\alpha_0 \beta}{\alpha \beta_0}\left(1 + \frac{\beta}{\beta_0}\right)^n\right]. \tag{199}$$

The computation of the residue at $z = 0$ is a bit more involved due to multiple order poles coming from the expansion of $\frac{1}{z}\left(1 + \frac{1}{\gamma z}\right)^n$. The residue at $z = 0$ is

$$\sum_{l=0}^n \binom{n}{l}\frac{1}{\gamma^l}\frac{1}{(l+1)!}\frac{d^{l+1}}{dz^{l+1}}\left.\frac{z^2 - 1}{(z - \delta)(1 - z\delta)}\right|_{z=0}. \tag{200}$$

The $l + 1$ order derivative can be computed using the simple elements decomposition

$$\frac{1}{(z - \delta)(1 - z\delta)} = \frac{1}{1 - \delta^2}\left(\frac{1}{z - \delta} - \frac{1}{z - \frac{1}{\delta}}\right) \tag{201}$$

and the fact that

$$\frac{1}{(l+1)!}\frac{d^{l+1}}{dz^{l+1}}\left.\frac{z^2 - 1}{z - z_0}\right|_{z=0} = \frac{1}{z_0^l}\left(\frac{1}{z_0^2} - 1\right). \tag{202}$$

Gathering all terms together yields the following expression of the residue at $z = 0$

$$\frac{1}{2}\left[\left(1 + \frac{\alpha}{\alpha_0}\right)^n + \frac{\alpha_0 \beta}{\alpha \beta_0}\left(1 + \frac{\beta}{\beta_0}\right)^n\right], \tag{203}$$

which implies that the contour integral in (198) is equal to $\left(1 + \frac{\alpha}{\alpha_0}\right)^n$ and concludes the proof of the formula (191).

## C.3 Computation of the densities

The mean particle density associated to the species $\epsilon = \pm$ at site $i$ is defined by the probability to observe a particle of species $\epsilon$ at site $i$ in the stationary state. We need to average over the two steps (odd and even) of the dynamics: in the stationary state the system is half of the time described by the probability vector $p$ and the other half by the probability vector $p'$. The density at site $i = 2k + 1$ is thus given by

$$n_\epsilon(i) = \frac{1}{2Z_L}\left(\sum_{\tau\in\{0,+,-\}}\langle l|_\tau\right)T^{k-1}(V_0 + V_+ + V_-)W_\epsilon T^{\frac{L-3}{2}-k}\left(\sum_{\tau,\tau'\in\{0,+,-\}}|rr\rangle_{\tau\tau'}\right)$$
$$+\frac{1}{2\tilde{Z}_L}\left(\sum_{\tau,\tau'\in\{0,+,-\}}\langle ll|_{\tau\tau'}\right)T^{k-1}V_\epsilon(W_0 + W_+ + W_-)T^{\frac{L-3}{2}-k}\left(\sum_{\tau\in\{0,+,-\}}|r\rangle_\tau\right),$$
(204)

where $T = (W_0 + W_+ + W_-)(V_0 + V_+ + V_-)$ and $\tilde{Z}_L$ is the normalisation of $p'$. Using the explicit expression of the matrices (93) and boundary vectors (94), (95), (102) and (101) we can use very similar techniques that have been used to compute the particle currents (partial action of the coherent states and the closure relation) to show that

$$\frac{1}{Z_L}\left(\sum_{\tau\in\{0,+,-\}}\langle l|_\tau\right)T^{k-1}(V_0 + V_+ + V_-)W_\epsilon T^{\frac{L-3}{2}-k}\left(\sum_{\tau,\tau'\in\{0,+,-\}}|rr\rangle_{\tau\tau'}\right)$$
$$=\frac{\alpha}{\alpha+\alpha_0}\frac{\beta_0\alpha_\epsilon\left(1+\frac{\alpha}{\alpha_0}\right)^{L-2}-\alpha_0\beta_\epsilon\left(1+\frac{\beta}{\beta_0}\right)^{L-2}}{\beta_0\alpha\left(1+\frac{\alpha}{\alpha_0}\right)^{L-2}-\alpha_0\beta\left(1+\frac{\beta}{\beta_0}\right)^{L-2}}$$
$$+\frac{(\alpha\beta_\epsilon-\beta\alpha_\epsilon)\left(1+\frac{\alpha}{\alpha_0}\right)^{i-2}\left(1+\frac{\beta}{\beta_0}\right)^{L-i-1}}{\beta_0\alpha\left(1+\frac{\alpha}{\alpha_0}\right)^{L-2}-\alpha_0\beta\left(1+\frac{\beta}{\beta_0}\right)^{L-2}}$$
(205)

and

$$\frac{1}{\tilde{Z}_L}\left(\sum_{\tau,\tau'\in\{0,+,-\}}\langle ll|_{\tau\tau'}\right)T^{k-1}V_\epsilon(W_0 + W_+ + W_-)T^{\frac{L-3}{2}-k}\left(\sum_{\tau\in\{0,+,-\}}|r\rangle_\tau\right)$$
$$=\frac{\beta}{\beta+\beta_0}\frac{\beta_0\alpha_\epsilon\left(1+\frac{\alpha}{\alpha_0}\right)^{L-2}-\alpha_0\beta_\epsilon\left(1+\frac{\beta}{\beta_0}\right)^{L-2}}{\beta_0\alpha\left(1+\frac{\alpha}{\alpha_0}\right)^{L-2}-\alpha_0\beta\left(1+\frac{\beta}{\beta_0}\right)^{L-2}}$$
$$+\frac{(\alpha\beta_\epsilon-\beta\alpha_\epsilon)\left(1+\frac{\alpha}{\alpha_0}\right)^{i-2}\left(1+\frac{\beta}{\beta_0}\right)^{L-i-1}}{\beta_0\alpha\left(1+\frac{\alpha}{\alpha_0}\right)^{L-2}-\alpha_0\beta\left(1+\frac{\beta}{\beta_0}\right)^{L-2}}.$$
(206)

Summing both contribution we end up with

$$n_\epsilon(i) = \frac{1}{2}\left(\frac{\alpha}{\alpha+\alpha_0}+\frac{\beta}{\beta+\beta_0}\right)\frac{\beta_0\alpha_\epsilon\left(1+\frac{\alpha}{\alpha_0}\right)^{L-2}-\alpha_0\beta_\epsilon\left(1+\frac{\beta}{\beta_0}\right)^{L-2}}{\beta_0\alpha\left(1+\frac{\alpha}{\alpha_0}\right)^{L-2}-\alpha_0\beta\left(1+\frac{\beta}{\beta_0}\right)^{L-2}}$$
$$+\frac{(\alpha\beta_\epsilon-\beta\alpha_\epsilon)\left(1+\frac{\alpha}{\alpha_0}\right)^{i-2}\left(1+\frac{\beta}{\beta_0}\right)^{L-i-1}}{\beta_0\alpha\left(1+\frac{\alpha}{\alpha_0}\right)^{L-2}-\alpha_0\beta\left(1+\frac{\beta}{\beta_0}\right)^{L-2}}.$$
(207)

Taking into account that $\alpha + \alpha_0 = 1$ and $\beta + \beta_0 = 1$, the expression reduces to

$$n_\epsilon(i) = \frac{1}{2}(\alpha+\beta)\frac{\alpha_\epsilon(1-\beta)^{L-1}-\beta_\epsilon(1-\alpha)^{L-1}}{\alpha(1-\beta)^{L-1}-\beta(1-\alpha)^{L-1}}+\frac{(\alpha\beta_\epsilon-\beta\alpha_\epsilon)(1-\alpha)^{L-i}(1-\beta)^{i-1}}{\alpha(1-\beta)^{L-1}-\beta(1-\alpha)^{L-1}}.$$
(208)

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
