# Peer review of "Two-species hardcore reversible cellular automaton: matrix ansatz for dynamics and nonequilibrium stationary state"

_SciPost Physics, doi:SciPost Phys. 6, 074 (2019)_

## Round 1 · Referee Report · Anonymous (Referee 1) · 2019-5-1

Strengths

1- Derives exact results for out-of-equilibrium transport 2- Simple approach based on matrix product ansatz, independent of the Bethe ansatz / generalized hydrodynamics technology 3- Rich physics, with features of both ballistic and diffusive transport

Weaknesses

1- The considered model has a deterministic dynamics, which might miss some aspects of interacting quantum integrable models 2- Writing and grammar should be improved

Report

The authors introduce a one-dimensional two-species model with deterministic dynamics, driven out of equilibrium from three different protocols : a local quench at the origin, an inhomogeneous quench corresponding to joining two halves of the system each in a different stationary state, and boundary driving.
The time evolution of the density profiles (or, respectively, their stationary distribution in the case of boundary driving) are determined analytically using a matrix product ansatz. From there the scaling limit (large time, long distances) is obtained by recasting the analytical expressions by contour integrals and evaluating them at their saddle point. This allows for the observation of non-trivial transport features, among which a diffusive dynamics and the propagation of ballistic fronts.

The issue of transport, whether ballistic, diffusive or subdiffusive in many-body systems has attracted a lot of attention over the last years. In this perspective, the present works is remarkable in that it derives exactly some interesting features of transport, without relying on the assumptions of generalized hydrodynamics. It therefore comes as an interesting alternative to the latter.

Some criticism can still be adressed. It may be argued that models such as the one considered here may not catch in full generality the richness of interacting quantum many-body system: the dynamics here is indeed purely deterministic, and it would be interesting if the authors could discuss a little what it might miss compared to non-deterministic models. For the latter, it seems reasonable to expect that no such simple matrix product solution could exist.
Last but not least, there is some room for improvement in the grammar and writing (see details in next paragraph).

Once these aspects and the one addressed below are taken in consideration, i recommend publication in JSTAT.

Requested changes

1- Many syntax or spelling errors, which need to be corrected. For instance, 2nd paragraph of first page, misspell of "analytically". Later, "in the context of ["the" missing] Totally Asymmetric Simple Exclusion Process". 3rd paragraph, "as a propagation of ["the" to be replaced by "a"] local observable". 3rd paragraph of section 2.1, "usefull" should be replaced by "useful", etc... In many other places the writing seems a bit careless, and should be improved before publication.

2- In the introduction, since the TASEP is mentioned, it would be useful if the authors could recall in a few sentences the state of the art for the latter, as far as the computation of out-of-equilibrium density profiles is concerned.

3- The authors refer to a different approach to cellular automata, based on soliton counting. The deterministic dynamics considered here looks like it could allowed such methods for the present problems. Have these been considered by the authors? Do they fail where the matrix product approach works ? If so, this should probably be explained in the introduction.

4-After equation (6) the author mention a connection with the Yang-Baxter equation, and "the fact that $R(0)=U$ implies the integrability of the model with periodic boundary conditions.". This Last sentence is in fact not completely clear to me : while integrable R matrices can indeed be used to build commuting sets of transfer matrices in different geometries (row-to-row, or diagonal-to-diagonal), it is not clear whether the generator $\mathbb{U}$ for the dynamics considered here really belongs to this setup. It would correspond to a diagonal-to-diagonal array of R matrices all carrying an infinite spectral parameter, which seems difficult to relate to the usual construction. Could the authors clarify whether the Yang-Baxter integrability plays any role here ?

5- in eqs (11) and (12), $p'$ has not been defined.

6- eq (13) is slightly confusing, as the labels $\alpha_{0,\pm}, \beta_{0,\pm}$ have already been used in eq. (10), but are a priori unrelated (even though in the particular case of boundary driving they turn out to be the same). Could this be fixed ?

7- The notation $\langle +0 \rangle_{12}$ (and other similar ones) should be explained.

8- before equuation (22) : "On [remove "the"] physical grounds"

9- section 2.2 : Could the authors briefly discuss what happens when the condition (23) is not met ? Does the shock smoothen and decay ?

10- section 3.3, after (62) : maybe a few words to explain what a "marginally stable" shock means in practice.

11- Section 4.1, after (64) : "This dependence is reminiscent [of] the defect..."

12.- Figure 4 : is there a specific reason why the authors chose $\rho_+ = 0.01$, and not $0$ ? Would the plots look much different there ?

---

## Round 1 · Referee Report · Anonymous (Referee 2) · 2019-5-9

Strengths

exact analytical results for interacting system
good interpretation with respect to general out-of-equilibrium theory
timely and subject of high current interest

Weaknesses

the system is very particular, universality of results / principles not clear yet

Report

In this paper, the authors study a new model of cellular automata, analysing various out-of-equilibrium situations. Analytic results are obtained both for the exact states emerging and for averages of various observables, which are accurate and rigorous. These are compares with explicit numerical evaluations, showing good agreement. The hydrodynamic limit of large space and time is taken, and interpreted within standard hydrodynamic theory. The model is a model for two species of interacting particles with hard core interaction.

The paper is precise, with exact analytical expressions for interesting physical quantities, and fits very well within the modern quests in non-equilibrium physics. A strength is that it combines analytical accuracy and physically interesting questions. I gives additional information that is of great use in deciphering non-equilibrium phenomena. I recommend publication, but I’d like to see one aspect clarified.

This is about the hydrodynamic interpretation. On page 8, the standard shock analysis is performed, solving the Rankine-Hugoniot condition for the two-component fluid, and this is used to analyse shocks between separable states. The condition of equality of shock speed for the components (eq 22) is said to be on physical grounds. This is not so clear. In general, there is no problem in having many shocks at different velocities, and the Riemann problem for two-component fluids will usually display a shock and a rarefaction wave, or modifications of such two structures. I think what the authors are really doing here is establishing the conditions for a single shock to separate two current-carrying state.

This is then used later, pages 14-16, to interpret the result. But there are some questions. Wouldn’t it be possible to interpret all discontinuities in figure 3 using hydrodynamics? The sharp ones due to the light cone surely also have an interpretation; are they the normal-mode velocities associated to the vacancy propagation? So these would be ``contact singularities”, and the associated modes would be essentially free (and of which there are two due to the odd-even separation), and freeness then explaining why they would not display diffusive spreading? While the middle one, with diffusive spreading, isn’t it also a contact discontinuity (i.e. a discontinuity parallel to the mode’s velocity - perhaps this is just a question of words, here referred to as marginally stable)? Overall, the fluid has three components. So don’t we indeed have in general three normal modes, and aren’t these the three velocities that we see emerging?

Finally, my understanding is that on both sides of the diffusive discontinuity, the authors expect the state to be essentially one of the current-carrying states studied around eq 22. Can they see this analytically?

With all this, one would indeed have a standard picture for the Riemann problem of a 3-component fluid, with 3 structures (here discontinuities), and in the present particular case, all being contact discontinuity (which is actually unusual, and point probably to the integrability of the model).

Requested changes

add discussion elements about hydrodynamic interpretation (see report), especially of first case studied (but also of other cases if possible), in order to possibly better extract general principles.

---

## Round 2 · Referee Report · Anonymous (Referee 1) · 2019-5-22

Report

I thank the authors for their careful consideration of the points raised in my previous report, and am now fully satisfied with the present version.

---

## Round 2 · Referee Report · Anonymous (Referee 2) · 2019-6-5

Report

I thank the referees for clarifying the hydrodynamic properties of their results - this is very useful.

---

## Round 2 · List of Changes

Warnings issued while processing user-supplied markup:

  • Inconsistency: plain/Markdown and reStructuredText syntaxes are mixed. Markdown will be used.
    Add "#coerce:reST" or "#coerce:plain" as the first line of your text to force reStructuredText or no markup.
    You may also contact the helpdesk if the formatting is incorrect and you are unable to edit your text.

REFEREE 1:

We thank the referee for her/his careful review of our manuscript. We have revised our paper according to his/her remarks, and the remarks of the second referee.

ANSWER TO SPECIFIC REFEREE's comments:

Referee: Some criticism can still be addressed. It may be argued that models such as the one considered here may not catch in full generality the richness of interacting quantum many-body system: the dynamics here is indeed purely deterministic, and it would be interesting if the authors could discuss a little what it might miss compared to non-deterministic models. For the latter, it seems reasonable to expect that no such simple matrix product solution could exist.

Answer: This is an excellent point. And indeed, one can think of a stochastic deformation of our deterministic model which still keeps part of its ballistic character, and still probably can be exactly solved. At least we have empirical evidence that matrix product solutions of similar complexity should exist there, but that should require a non-negligible amount of extra work. This stochastic extension of the model and empirical evidence on its solvability are now discussed in a new paragraph at the end of Discussion and conclusion section.

  1. Referee: Many syntax or spelling errors, which need to be corrected. For instance, 2nd paragraph of first page, misspell of "analytically". Later, "in the context of ["the" missing] Totally Asymmetric Simple Exclusion Process". 3rd paragraph, "as a propagation of ["the" to be replaced by "a"] local observable". 3rd paragraph of section 2.1, "usefull" should be replaced by "useful", etc... In many other places the writing seems a bit careless, and should be improved before publication.

Answer: We have polished the spelling and grammar.

  1. Referee: In the introduction, since the TASEP is mentioned, it would be useful if the authors could recall in a few sentences the state of the art for the latter, as far as the computation of out-of-equilibrium density profiles is concerned.

Answer: We do not wish to specifically emphasise one stochastic model, even though TASEP is perhaps the most important one, but rather put the emphasis on the matrix product ansatz method, as our context (that of deterministic lattice gasses) is a very different one than TASEP. Therefore we feel that the mention in the introduction should be adequate.

  1. Referee: The authors refer to a different approach to cellular automata, based on soliton counting. The deterministic dynamics considered here looks like it could allowed such methods for the present problems. Have these been considered by the authors? Do they fail where the matrix product approach works ? If so, this should probably be explained in the introduction.

Answer: Our matrix product ansatz could be equivalently interpreted through "soliton counting" mechanism, but we feel that such a solution would look more convoluted and less clear than the one we discuss here, which is obtained through algebraic cancellation mechanism and its explicit representation in terms of Fock spaces. We quote in this context our recent independent work on Rule 54 cellular automaton, where we could only obtain matrix product solution for dynamics in terms of soliton counting and algebraic cancellation mechanism is still unknown.

  1. Referee: After equation (6) the author mention a connection with the Yang-Baxter equation, and "the fact that R(0)=U implies the integrability of the model with periodic boundary conditions.". This Last sentence is in fact not completely clear to me : while integrable R matrices can indeed be used to build commuting sets of transfer matrices in different geometries (row-to-row, or diagonal-to-diagonal), it is not clear whether the generator U for the dynamics considered here really belongs to this setup. It would correspond to a diagonal-to-diagonal array of R matrices all carrying an infinite spectral parameter, which seems difficult to relate to the usual construction. Could the authors clarify whether the Yang-Baxter integrability plays any role here ?

Answer: We have added a new paragraph after Eq. (7) (braid relation for U), explicitly defining the spectral-parameter dependent R-matrix obeying the Yang-Baxter equation, and constructing a commuting set of transfer matrices, through which we can derive the full evolution operator, and if needed, an infinite set of conserved quantities of the dynamics. This clearly demonstrated the relevance of integrability for the present model.

  1. Referee: in eqs (11) and (12), p' has not been defined.

Answer: We now explicitly define p and p' just after Eq. (15) (previous Eq. (12)).

  1. Referee: eq (13) is slightly confusing, as the labels α0,±,β0,± have already been used in eq. (10), but are a priori unrelated (even though in the particular case of boundary driving they turn out to be the same). Could this be fixed ?

Answer: We believe the notation is fully consistent as it is.

  1. Referee: The notation ⟨+0⟩12 (and other similar ones) should be explained.

Answer: A sentence has been added after Eq. (18) to explicitly explain this notation.

  1. Referee: before equation (22) : "On [remove "the"] physical grounds"

Answer: This has been rewritten.

  1. Referee: section 2.2 : Could the authors briefly discuss what happens when the condition (23) is not met ? Does the shock smoothen and decay ? Answer: see after question 10.

  2. Referee: section 3.3, after (62) : maybe a few words to explain what a "marginally stable" shock means in practice

Answer to 10 and 11: A new paragraph is added before the section 2.3, explaining in detail the notion of shock stability.

  1. Referee: Section 4.1, after (64) : "This dependence is reminiscent [of] the defect..."

Answer: Done

  1. Referee: Figure 4 : is there a specific reason why the authors chose ρ+=0.01, and not 0? Would the plots look much different there?

Answer: For ρ+=0, we cannot observe the balistically left moving peak on the density of + particles. The height of the peak is indeed proportional to ρ+, see eq. (89) (previously eq (87)). The fine-tuning of ρ+ to 0.01 was only done for graphical purpose.

REFEREE 2:

We thank the referee for her/his insightful remarks concerning the hydrodynamic aspect of the paper. In fact, her/his remarks has encouraged us to give more details and improve the presentation.

ANSWER TO SPECIFIC REFEREE's comments

  1. Referee: there is no problem in having many shocks at different velocities, and the Riemann problem for two-component fluids will usually display a shock and a rarefaction wave, or modifications of such two structures. I think what the authors are really doing here is establishing the conditions for a single shock to separate two current-carrying state.I think what the authors are really doing here is establishing the conditions for a single shock to separate two current-carrying state.

Answer: We agree with the referee: Indeed, several scenarios are possible hydrodynamically (formation of two spatially separated distinct shocks, a combination of a shock and a rarefaction wave, etc.), if two current carrying states are joined together. Our statements are now made more precise in the revised version, see the paragraph following the Eq (19) and a paragraph following the Eq (21).

  1. Referee: Overall, the fluid has three components. So don't we indeed have in general three normal modes, and aren't these the three velocities that we see emerging?

Answer: With respect to the remark of the referee, that the fluid has three components, we would like to note that only 2 quantities out of three are independent, since the sum of local densities of pluses, minuses and vacancies is constant, i.e. $n_{k,+}+n_{k,-}+n_{k,0}=1$ for all sites $k$.

So generically, in absence of odd/even separation, we have two conservation laws and one would expect two characteristic velocities. However, due to odd-even sublattice structure of the evolution operator, we find three characteristic velocities: one associated with the diffusive mode, (which can be seen e.g. in Fig.4), and two "free" modes, associated with a propagation of contact discontinuity Eq.(19),(20), which indeed result in being normal modes of vacancy propagation. In Fig.4 one sees a diffusive mode and one of the free modes (another free mode would appear, if one considers slightly more general initial condition with a local quench in two neighbouring (even and odd) sites).

  1. Referee: my understanding is that on both sides of the diffusive discontinuity, the authors expect the state to be essentially one of the current-carrying states studied around eq 22. Can they see this analytically?

Answer: The point discontinuities in Fig.3, in accordance with the suggestion of the referee, do belong to the contact singularities described in "hydrodynamic" section Eqs(19),(20). In the revised version, we have added a paragraph between Eqs. (19) and (20), and three paragraphs after Eq. (64), following Fig. 3, where we now discuss hydrodynamic interpretation of our microscopic results, summarized in Eqs. (60,61), in detail.

---

## Editorial Decision

published